# Recombinant protein platform for high-throughput investigation of peptide-liposome interactions via fluorescence anisotropy depolarization

Antonis Margaritakis[1], Meirui Qian[2], David H. Johnson [3], Wade F. Zeno [3], Tobias S. Ulmer [2] & Peter J. Chung [1,4,5] ✉

Many cytosolic proteins critical to membrane trafficking and function contain an unstructured domain that can bind to specific membranes, with a transition into an amphipathic helix induced upon membrane association. These inducible amphipathic helices often play a critical role in organelle recognition and subsequent function, but the tools used to characterize affinity towards specific membranes are low-throughput and dependent on the solubility of the inducible amphipathic helix. Here, we introduce a modular recombinant protein platform for rapidly measuring the binding affinity of inducible amphipathic helices towards a variety of membrane compositions and curvatures using high-throughput fluorescence anisotropy measurements. Inducible amphipathic helices are solubilized with a fluorescently-tagged small ubiquitin-like modifier (SUMO) protein and binding to membranes quantified by leveraging the unexpected decrease in fluorescence anisotropy upon binding, a phenomenon previously observed but not well understood. By using fluorescence anisotropy decay measurements, solution NMR experiments, and solution FRET, we deduce that this phenomenon likely occurs due to the local increase in fluorophore motion upon binding to the membrane enabled by vesicle membrane charge under low-salt conditions. This recombinant protein platform can be readily applied to any inducible amphipathic helix of interest, allowing for investigation of specific membrane biochemical parameters facilitating binding.

Inducible amphipathic helices are protein motifs that transition from a disordered state in solution to an amphipathic helix upon association with a particular membrane. These motifs are often found in peripheral membrane proteins related to membrane remodeling, signaling, and trafficking[1–7], with their peptide sequence driving their ability to recognize distinct membranes. For example, helix 0 of Amphiphysin (part of the NBAR superfamily of proteins) directly influences the protein's ability to associate with membranes, with its presence linked to reduced curvature sensitivity and weaker binding[8–10]. The N-terminal domain of the Endophilin family of proteins has an increased affinity for lipids found in the mitochondria and endosomes, consistent with its putative biological function[11–13]. Similarly, the N-terminal domain of Huntingtin binds preferentially to membranes with

high anionic lipid content and low cholesterol, mimicking mitochondrial membranes[14]. The ability of the inducible amphipathic helix to detect and bind to the correct cognate membrane is critical for proper function of the parent protein[9,15].

Although inducible amphipathic helices are crucial for subcellular localization, systematically identifying the parameters that govern their lipid targeting remains experimentally challenging[16]. First, their ability to bind to and insert into membranes necessitates a degree of hydrophobicity that, without its linkage to the necessarily more soluble parent protein, leads to instability in solution[17]. For example, some isolated N-terminal domains of the endophilin isoforms exhibit poor solubility and readily precipitate out of solution[13,18]. Furthermore, common techniques for extracting binding

[1]Department of Physics and Astronomy, University of Southern California, Los Angeles, CA, USA. [2]Department of Physiology and Neuroscience, Zilkha Neurogenetic Institute, Keck School of Medicine, University of Southern California, Los Angeles, CA, USA. [3]Mork Family Department of Chemical Engineering and Materials Science, University of Southern California, Los Angeles, CA, USA. [4]Department of Chemistry, University of Southern California, Los Angeles, CA, USA. [5]Alfred E. Mann Department of Biomedical Engineering, University of Southern California, Los Angeles, CA, USA. ✉e-mail: pjchung@usc.edu

isotherms (such as tryptophan fluorescence, isothermal titration calorimetry, circular dichroism, etc.) are inherently low throughput and sample intensive, with a single isotherm for one lipid composition potentially requiring hours to measure and large amounts of protein and lipid. Additionally, the sheer diversity of lipid species within trafficking organelles creates an enormous compositional phase space, making comprehensive exploration of the determinants of selective membrane binding a daunting experimental challenge[19]. Understanding the underlying parameters that govern this binding behavior demands a highly parallelized and systematic approach, as comparisons across studies of individual proteins are often challenging and often lead to contradicting conclusions. For instance, both the curvature inducing abilities (or absence thereof) of the helix 0 of Amphiphysin[11,18,20,21] and the role of the helix 0 of N-BAR Endophilin in membrane curvature generation[18,22,23] have been difficult to determine.

Here, we introduce a modular recombinant protein platform for high-throughput quantification of the binding affinity of inducible amphipathic helices to model membranes using fluorescence anisotropy measurements, commonly available in microplate readers. Our system solubilizes the peptide of interest using a C-terminally linked SUMO (Small Ubiquitin-like Modifier) protein (Uniprot accession #: Q12306). Not only does the SUMO protein helps solubilize our peptide of interest, it greatly amplifies expression yields in E.coli[24]. Moreover, our recombinant platform construct is co-expressed in an N-terminal acetylation system to better reflect modifications found in their native environments[25]. A fluorescent label (unless otherwise stated, an Oregon Green 488 conjugated through a maleimide reaction to a cysteine mutated into the C-terminal domain) allows for spectroscopic

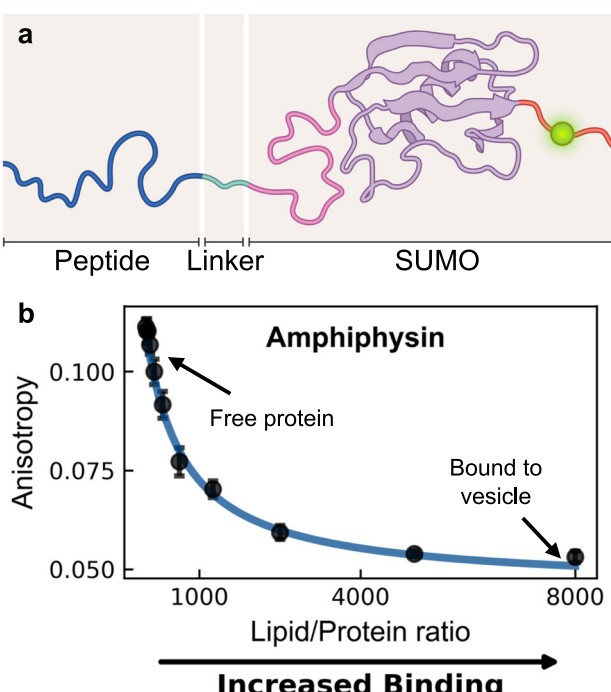

**Fig. 1 | Our recombinant protein platform exhibits a decrease in fluorescence anisotropy upon binding to a vesicle. a** Schematic representation of the SUMO protein platform. Peptides of interest (dark blue) are engineered at the N-terminus, linked to SUMO via soluble linker (light blue) and a mutated cysteine to facilitate fluorophore attachment (green). A cleavable C-terminal hexahistidine tag (not depicted) facilitates purification and cleaved prior to usage. Schematic represents individual engineered protein domains, not a conformation of the construct's free state. **b** An unexpected decrease in fluorescence anisotropy is observed when measuring the N-terminal domain of Amphiphysin engineering to our platform upon increasing titration of 67 nm diameter DOPC/DOPS (70/30) vesicles. Data was fit using a membrane partitioning model (blue) as described in Methods. Data are averages of $n = 3$ replicate wells with standard deviations plotted (error bars).

quantification of the peptide-liposome interaction via fluorescence anisotropy (Fig. 1a) and a downstream TEV-cleavable hexahistidine tag facilitates purification. Strikingly, we detect a decrease in fluorescence anisotropy upon binding (Fig. 1b). We expected the apparent larger size of the bound construct/vesicle system (upon inducible amphipathic helix binding) to lead to elevated fluorescence anisotropy, consistent with the slower rotational diffusion of the bound state[26]. The observed anisotropy decrease indicates a counterintuitive increase in rotational diffusion of our recombinant construct when bound to the much larger vesicle. However, given the clear sigmoidal characteristic of the anisotropy signal at higher lipid to protein ratios (albeit, in an unexpected direction), we continued to investigate the binding behavior using a complementary technique, tryptophan fluorescence. Surprisingly, we found strong quantitative agreement between our fluorescence anisotropy and tryptophan fluorescence measurements, suggesting that the decrease in fluorescence anisotropy corresponded to the bound state of our recombinant platform. To illuminate the mechanism that leads to this phenomenon, we investigated the dynamic structure of our recombinant platform using a combination of fluorescence anisotropy decay, nuclear magnetic resonance measurements, and solution fluorescence resonance energy transfer (FRET) measurements. These techniques reveal the presence of secondary, charge-induced interactions between part of the folded domain of SUMO and the negatively charged fluorophore that affect the conformation of the system both in the free and vesicle-bound states. Upon vesicle binding, the fluorescent label experiences varying degrees of secondary conformational freedom dependent on the amount of "visible" membrane charge, yielding anisotropies lower than those of the free state. When membrane charge decreases, or the environmental salt concentration increases, the dynamic range in fluorescence anisotropy (i.e., the difference between the free and fully bound states) is reduced, limiting the accuracy at which binding efficiencies can be determined. To assess whether there are electrostatically-mediated transient interactions between the folded SUMO domain, vesicles, and fluorophore, we introduced a series of charge-altering mutations to the construct, minimizing electrostatic interactions and thereby increasing the dynamic range in elevated salt conditions, expanding the platform's potential application to the near physiological salt regime for negatively charged membranes or to near uncharged membranes in the low salt regime.

The robustness of this phenomenon allows for this platform to be readily applied as a high-throughput screening tool to determine the dynamic parameters governing peptide-membrane interactions. By systematically mapping the regions of membrane phase space where protein binding is maximized, our platform enables a rigorous investigation of the peptide code responsible for precise protein subcellular localization, paving the way towards reconciling peptide sequence with organelle specificity.

## Results and discussion
### High-throughput detection of peptide-liposome binding
To leverage the decrease in fluorescence anisotropy of our platform as a high-throughput probe for binding affinity, we constructed binding isotherms for two distinct peptides against a $4 \times 3$ array of distinct vesicles. Along one axis of the array, we increased 1,2-dioleoyl-*sn*-glycero-3-phosphoethanolamine (DOPE) content (against decreasing 1,2-dioleoyl-*sn*-glycero-3-phosphocholine [DOPC] background and constant 30% 1,2-dioleoyl-*sn*-glycero-3-phospho-L-serine [DOPS]). On the other axis, we increased membrane curvature by extruding vesicles with decreasing diameters (Fig. 2a). We chose DOPE as its smaller headgroup relative to DOPC and DOPS promotes surface defects that may allow for an easier amphipathic helix insertion[3,27]. Similarly, many inducible amphipathic helices preferably associate with highly curved surfaces[28] (or even promote high curvatures)[29,30], potentially due to a curvature induced bilayer asymmetry facilitating easier access to the hydrophobic acyl chain region.

We expressed and purified recombinant platform constructs presenting peptides corresponding to the inducible amphipathic helices (AH) of the CHMP4B (AA1-19, Uniprot: Q9H444) subunit from the ESCRTIII protein complex and Huntingtin (AA1-17, Uniprot: P42858), hereinafter referred to

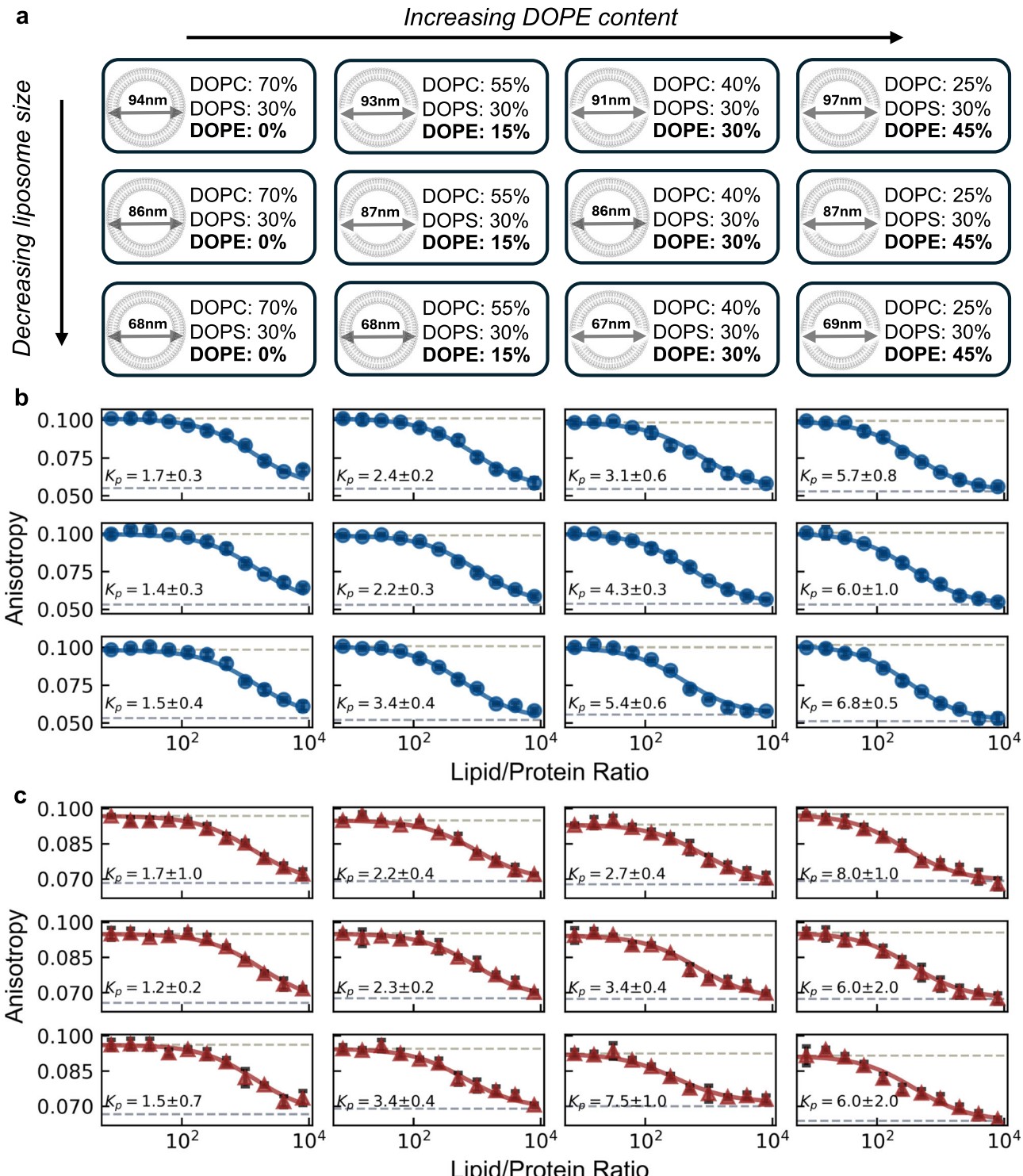

**Fig. 2 | High-throughput detection of inducible amphipathic helix binding to liposomes is possible. a** Schematic representation of the vesicle array used to assess protein binding in different membrane compositions and curvatures. Four vesicle composition batches were prepared: DOPC/DOPS/DOPE = (70−X)/30/X with X = (0, 15, 30, 45), each extruded to three different final diameters D ≈ (94 nm, 86 nm, 68 nm). Fluorescence anisotropy measurements of our recombinant platform engineered with the inducible amphipathic helices of **b** CHMP4B (AA:1–19, blue circles) and **c** Huntingtin (AA:1-17, red triangles) with increasing vesicle concentration. Data points show the mean of $n = 3$ replicate wells for each lipid/protein, with standard deviations plotted (black error bars). Data were fit using a membrane partitioning model (solid lines) with partition coefficient values shown as ×10[5].

as the CHMP4B-AH and Htt17-AH respectively. We constructed each binding isotherm by measuring the fluorescence anisotropy at 12 lipid/protein ratios of our 4 × 3 array. Each 100 μL sample had a protein concentration of 0.25 μM, with three replicate wells for each lipid/protein ratio. The resulting binding isotherms were fit to the partition equilibrium function assuming a single bound state (see Methods), yielding a corresponding partition coefficients (Fig. 2b,c, Fig. S1). The CHMP4B-AH exhibited a considerable preference for membranes with higher DOPE content across the range of membrane curvatures tested. Irrespective of curvature, the fitted partition coefficient increases ~5-fold for higher DOPE content. Interestingly, the

**Fig. 3 | Binding isotherms derived from fluorescence anisotropy and tryptophan fluorescence show strong quantitative agreement.** Parallel fluorescence anisotropy and tryptophan fluorescence measurements were taken with our recombinant platform engineering with the inducible amphipathic helices of **a, b** Amphiphysin (AA: 1-25, F9W mutant), **c, d** CHMP4B (AA: 1-19, F8W mutant), **e, f** Endophilin-B1 (AA: 1-33, F18W mutant) and **g, h** Huntingtin (AA: 1-17, F11W mutant). The binding of each platform was measured against changes in a variable lipid membrane content either known or suspected to influence binding affinity: cholesterol (Chol) for Amphiphysin, phosphatidylethanolamine (PE) for Endophilin-B1, cardiolipin (CL) for CHMP4B, and cardiolipin (CL) for Huntingtin. In all cases, vesicles were measured to be ~70 nm diameter, composed of DOPC/DOPS/variable lipid at (70-X)/30/X molar ratios. Anisotropy bound-ratio data are calculated from anisotropy averages of $n = 4$ measured wells with error bars showing corresponding standard deviations as described in Methods. For the fitted partition coefficient values, error bars represent the standard errors of each fit.

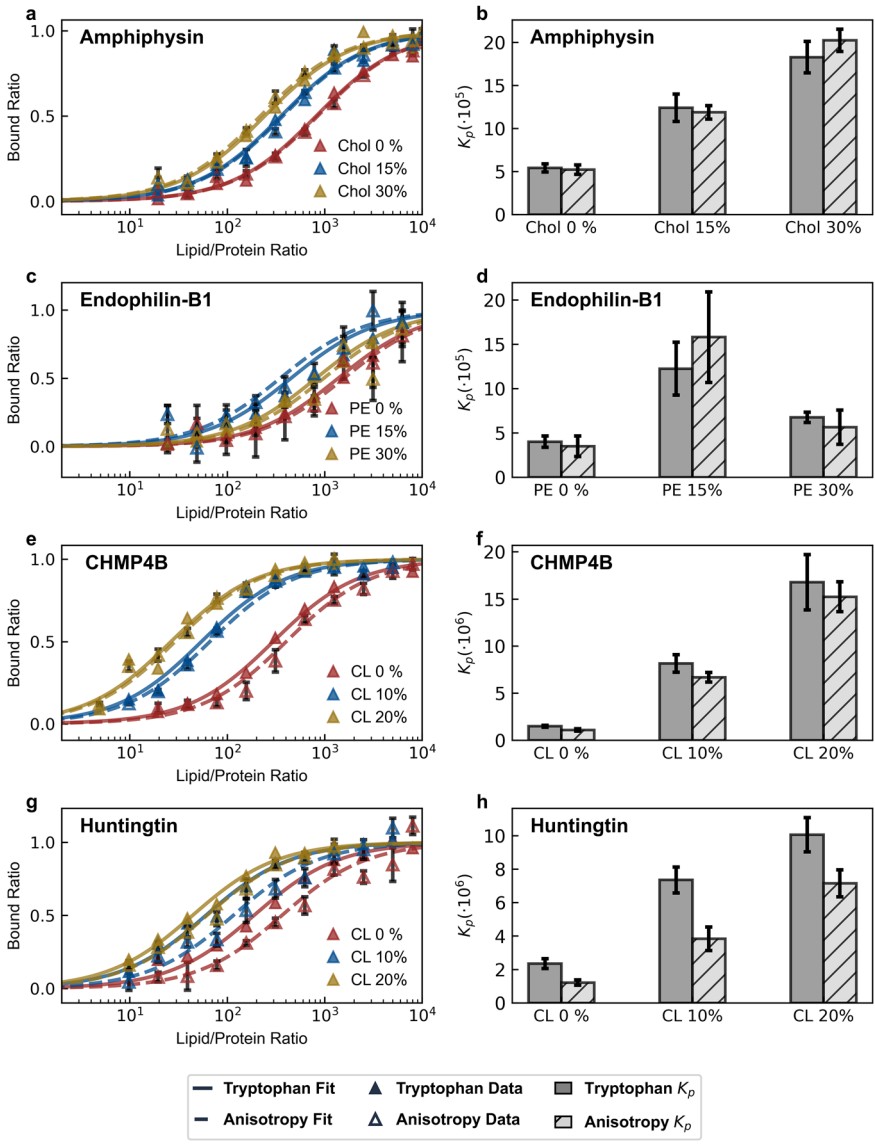

CHMP4B-AH exhibited a preference for higher membrane curvatures at higher DOPE content (30%, 45%), suggesting synergistic effects. A similar behavior is observed for the Htt17-AH, which showed increased partitioning towards membranes with higher DOPE content. Vesicles with 15% and 30% DOPE content appeared to facilitate the emergence of a curvature preference in binding, with 68 nm diameter vesicles exhibiting a significantly higher binding affinity than vesicles with diameters 86 nm and 94 nm.

**Tryptophan fluorescence measurements validate our recombinant platform as a tool for measuring binding affinity**
Having extracted partition coefficients from our recombinant platform in a high-throughput manner, we sought to rigorously compare our probe to values derived from a complementary technique. Although slower and much more sample-intensive, tryptophan fluorescence is considered a rigorous, quantitative technique which can evaluate the binding of inducible amphipathic helices with membranes[31–33]. It leverages the natural fluorescence of tryptophan which changes depending on the polarity of its immediate environment (with an emission centered ~350 nm in aqueous solvents and ~330 nm when submerged in the nonpolar acyl chain layer). Conveniently, all peptides tested contain at least one phenylalanine, which can be mutated into a physiochemically similar tryptophan, enabling parallel measurements using fluorescence anisotropy (Fig. S2) and tryptophan

fluorescence (Fig. S3). Although tryptophan has a higher hydrophobicity than phenylalanine, which is expected to somewhat alter the binding affinity of the inducible amphipathic helix, the overall effect should not yield a significantly different binding profile than the wildtype helix sequence[13,34,35].

We extracted binding isotherms of four distinct peptides to vesicles with systematically varied membrane compositions, specifically targeting compositions known to influence the binding affinity (DOPE and cardiolipin content for Endophilin-B1[3,13] and Huntingtin[3,14], respectively) or suspected of doing so (cholesterol and cardiolipin content for Amphiphysin[36,37] and CHMP4B[4,38], respectively) (Fig. 3). Binding isotherms showed strong quantitative agreement between fluorescence anisotropy and tryptophan fluorescence measurements, with consistent results across systematically varied membrane compositions. We also performed control measurements using a platform construct without conjugated peptides (SUMO-fluorophore) to confirm that the binding isotherms exclusively reflect interactions with the inducible amphipathic helices (Fig. 4).

During these measurements, we uncovered a previously unreported result. We detect a significant monotonic increase in partitioning for helix 0 of Amphiphysin towards vesicles containing increased cholesterol content. Helix 0 of Amphiphysin exhibits ~2-fold and ~4-fold higher partition coefficients for vesicles containing 15% and 30% cholesterol, respectively, compared to vesicles without cholesterol. In both cases, binding isotherms

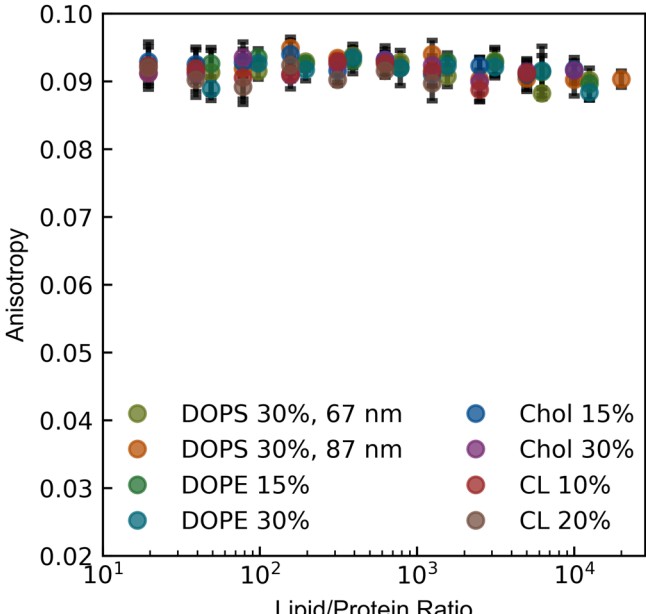

**Fig. 4 | Without conjugated peptides, our platform is inert to lipid vesicles.**
Fluorescence anisotropy measurements of fluorescently-labeled recombinant platform with no conjugated peptide with varying liposome compositions. In all cases, vesicles were measured to be ~70 nm in diameter unless otherwise noted, composed of DOPC/DOPS/variable lipid $(70-X)/30/X$. Variable lipids were DOPE, cardiolipin (CL) and cholesterol (Chol). Protein concentrations were 100–250 nM depending on experiment. Data points show mean of $n = 4$ replicate wells for each lipid/protein ratio, with standard deviations plotted (black error bars).

produced from the two techniques are in strong quantitative agreement (Fig. 3a,b).

In measuring the binding affinity of the CHMP4B-AH, both techniques exhibited clear preferential partitioning into membranes with higher cardiolipin content. A 10% cardiolipin membrane content in vesicles yields a 5-fold increase in partitioning relative to vesicles containing none, while 20% content yields a further ~3-fold increase. Notably, both techniques consistently captured the non-monotonic binding behavior of helix 0 of Endophilin-B1 in response to increasing DOPE concentrations (Fig. 3c–f). For vesicles composed of DOPC/DOPS (70/30) with an average diameter of ~68 nm, we calculated a partition coefficient of $(4 \pm 0.6) \times 10^5$, a result in close agreement with the value of $(1.5 \pm 0.5) \times 10^5$ reported by Robustelli and Baumgart for vesicles similarly comprised of DOPC/DOPS (75/25) with an average diameter of 100 nm[13].

Finally, we tested the Htt17-AH with vesicles containing increasing amounts of cardiolipin. Both techniques unveiled a strong preference for membranes with higher cardiolipin content. Specifically, tryptophan fluorescence showed ~2-fold and ~3-fold increases for vesicles containing 10% and 20% cardiolipin, respectively, compared to vesicles with no cardiolipin content. In contrast, our platform's fluorescence anisotropy decrease indicated ~3-fold and ~9-fold increases for the same compositions (Fig. 3g,h). While the magnitude of response differs between the two methods, both techniques consistently capture the same qualitative trend (namely, the preference of Htt17-AH for cardiolipin-containing membranes). We do not expect the binding isotherms between the two techniques to always be identical, as tryptophan fluorescence is a direct measure of binding while our platform's fluorescence anisotropy reflects the proximity of the platform construct to the membrane. Nevertheless, we confirmed binding via the formation of an amphipathic helix (for all peptides) using circular dichroism spectroscopy (Fig. S4). Our results clearly indicate that both methods can effectively discriminate the membrane compositional preferences of Htt17.

In all tested scenarios, the individual lipid preferences can be extracted from fluorescence anisotropy curves just as effectively as from tryptophan

fluorescence data. The strong quantitative agreement between resultant binding isotherms and extracted partition coefficients underscores the robustness of our recombinant protein platform as a precision tool for measuring the binding affinity of inducible amphipathic helices, prompting further investigation into the mechanism by which a decrease in fluorescence anisotropy is observed upon binding.

### The fluorescence anisotropy decrease is dependent on apparent membrane charge, but not size or choice of fluorophore

To elucidate the mechanism by which our recombinant platform exhibited a decrease in fluorescence anisotropy upon binding, we devised a model version of our platform to systematically test the membrane conditions that could drive this effect. To achieve this, we replaced the peptide corresponding to an inducible amphipathic helix with hexahistidine (His$_6$-SUMO-fluorophore). By incorporating a nickel-chelating lipid (10% 1,2-di-(9Z-octadecenoyl)-$sn$-glycero-3-[(N-(5-amino-1-carboxypentyl)iminodiacetic acid)succinyl]) into our vesicles, our model recombinant platform construct bound tightly to the membrane surface irrespective of specific peptide-vesicle interactions. This design enabled us to isolate the contribution of the membrane itself to the decrease in fluorescence anisotropy, independent of the unique binding behavior of each peptide.

Using this model construct, we dissected the contribution of the following physiochemical parameters: vesicle size, membrane charge, salt conditions, and fluorophore identity (Fig. 5, Fig. S5). These parameters were chosen as representative factors anticipated to most likely affect the local mobility of the negatively charged fluorophore (at physiological pH) within its immediate environment, potentially resulting in decreased fluorescence anisotropy when bound to the larger vesicle. The decrease in fluorescence anisotropy seemed to be independent of fluorophore and vesicle size (Fig. S5, Fig. 5a), yet highly dependent on membrane charge and salt concentration (Fig. 5b,c), indicating an electrostatically driven phenomenon. Specifically, the progressive narrowing of the dynamic detection window with increasing salt concentration mirrored the effect exhibited with decreasing membrane negative charge. To ensure binding occurs at low membrane charge conditions, microscopy measurements confirmed colocalization occurs with DOPC/DGS-NTA(Ni) vesicles (Fig. S6). Taken together, these results strongly suggest that the decrease in fluorescence anisotropy exhibited by the system is strongly coupled to the amount of membrane charge "visible" to the fluorophore conjugated to the C-terminal domain of our His$_6$-SUMO-fluorophore construct. Supporting this hypothesis, altering the pH of our sample solution led to a decrease in fluorescence anisotropy for the free state of all recombinant constructs tested, potentially due to pH induced charge changes in the fluorescently-labeled C-terminal domain (Fig. S7). To further test this idea, we introduced a charge-altering mutation into the fluorescently-tagged C-terminal domain (R104H). This mutation narrowed the dynamic detection window without affecting the overall binding isotherm (Fig. S8). Overall, the presence of some membrane charge detectable by the fluorophore is crucial for measurement via anisotropy decrease (Fig. 5c). Increasing NaCl concentration in the experimental buffer sharply increases the membrane charge requirement, with concentrations over 100 mM abolishing any measurement regardless of membrane charge. These findings suggest that the decrease in fluorescence anisotropy is directly linked to membrane charge exposure and indicates that local conformational changes within the fluorescently-tagged C-terminal domain can modulate this effect without affecting overall binding.

### A decrease in fluorescence anisotropy occurs upon increasing fluorophore mobility upon binding

To investigate the underlying conformational dynamics of our platform, we employed time-resolved fluorescence anisotropy decay. Unlike steady-state measurements, this technique delivers a precise excitation pulse of polarized light and monitors the time-dependent depolarization of the emitted fluorescence. Analyzing this decay over time allows us to separate the depolarization due to the potentially faster rotational diffusion of the

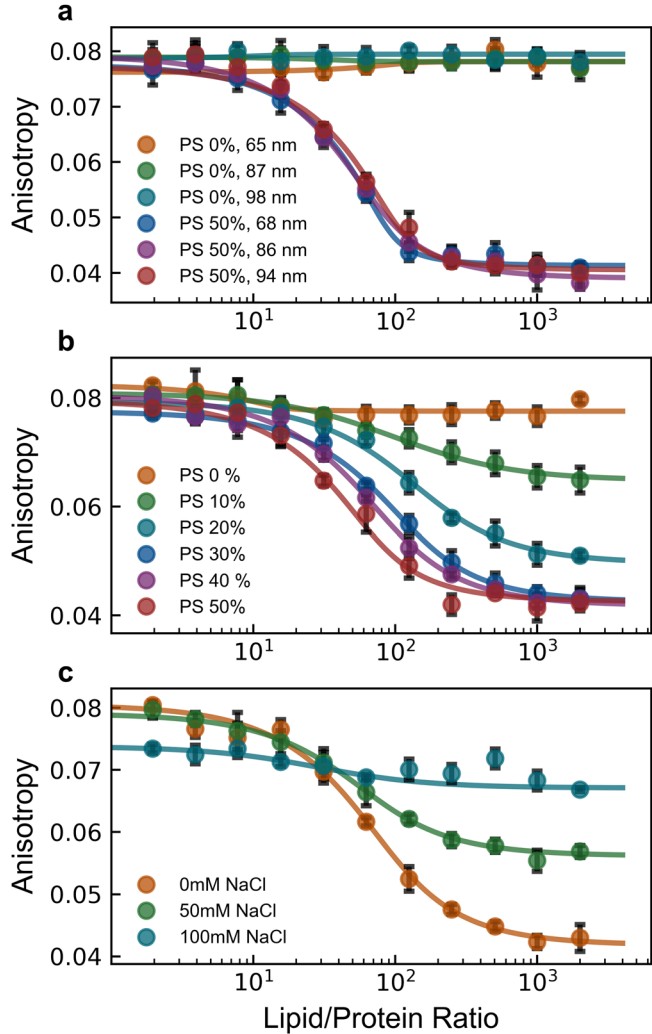

**Fig. 5 | The decrease in fluorescence anisotropy is electrostatic in nature.**
Fluorescence anisotropy was measured for fluorescently-tagged hexahistidine model platform binding to nickel-chelating lipid containing vesicles against different conditions: **a** decreasing vesicle size (with lipid vesicle compositions of DOPC/DGS-NTA(Ni) = 90/10 or DOPC/DOPS/DGS-NTA(Ni) = 50/40/10), **b** membrane charge (with lipid vesicle compositions of DOPC/DOPS/DGS-NTA(Ni) = (90-$X$)/$X$/10 with X shown in the legend), and **c** buffer salt concentration (with lipid vesicle compositions of DOPC/DOPS/DGS-NTA(Ni) = 50/40/10). Data points are the mean of $n = 4$ replicate wells for each lipid/protein ratio with standard deviations plotted (black error bars). Solid lines represent fits to a depletion model as described in methods. Unless otherwise noted, all vesicles were extruded to a final diameter of ~95 nm.

fluorophore from the slower diffusion of the platform construct or construct/vesicle complex, providing mechanistic insights into the observed changes in steady-state anisotropy upon binding. These experiments required decay measurements in three states: free in solution, bound to highly negatively charged vesicles (to recapitulate the observed steady-state effect), and bound to vesicles with low charge (which exhibits no change in anisotropy even though the protein is membrane-bound). We again used the hexahistidine model construct (His$_6$-SUMO-fluorophore) to examine the bound-state behavior. We first attempted to fit the decay of the three states with a model corresponding to a single rotational correlation lifetime, consistent with a fluorophore rigidly attached to the platform construct (in a free state) or the construct/vesicle complex (in the bound states). Unfortunately, this model fails to capture the early decay behavior of the fluorophore (Fig. S9), suggesting a significant contribution from local fluorophore motion. As a result, we explored using a model in which there is an additional hindered degree of freedom, representing a fluorophore that is non-rigidly attached to the platform construct. This two-state hindered anisotropy decay model better represents fluorophores attached to macromolecules with segmented mobility[39]:

$$r(t) = r_0\left(\alpha e^{-\frac{t}{\theta_F}} + (1 - \alpha)\right)e^{-\frac{t}{\theta_P}}$$

Where $\theta_F$ and $\theta_P$ represent the local and global rotational correlation times, respectively, and $\alpha$ is the fractional amplitude of the fast (local) motion in relation to the slower (global) motion. This model provides a better fit to the decay data, as indicated by smaller residuals and the absence of systematic deviations from baseline (Fig. 6a, Fig. S10a–f). For the free state, $\theta_P$ was determined to be $(3 \pm 1)$ ns, closely matching the expected value of ~4 ns calculated using the Stokes–Einstein–Debye equation. Samples containing vesicles yielded fitted parameters indicating a large global rotational correlation lifetime ($\theta_P$), several orders of magnitude greater than the local fluorophore rotational correlation lifetime ($\theta_F$) (Table S1). In this case, the fitted value of $\theta_P$ approaches infinity and indicated that the rotational diffusion is dominated by the vesicle, which has a calculated correlation time of ~100 µs. As a result, early time anisotropy decay is governed primarily by the local motion of the fluorophore, likely arising from its attachment to the disordered C-terminal domain of our His$_6$-SUMO-fluorophore construct. The fast local fluorophore motion, with a fitted $\theta_F \sim 1$ ns (and significantly shorter than the fluorophore fluorescence lifetime of ~4 ns), rapidly depolarizes the emitted photons and dominates the overall decay. We can further refine our bound state model to account for the slow platform construct/vesicle coupled motion by modifying the two-state decay in the limit where $\theta_P \rightarrow \infty$:

$$r(t) = r_0\left(\alpha e^{-\frac{t}{\theta_F}} + (1 - \alpha)\right) = (r_0 - r_\infty)e^{-\frac{t}{\theta_F}} + r_\infty$$

**Fig. 6 | Fluorescence anisotropy decay supports a model by which a decrease in fluorescence anisotropy occurs due to a local increase in fluorophore mobility.** Fluorescence anisotropy decay was measured for fluorescently-tagged hexahistidine model platform (25 nM) in three different states: **a** free in solution, **b** bound to neutral vesicles (DOPC/DGS-NTA(Ni) = 90/10, 25 µM) and **c** bound to negatively charged vesicles (DOPC/DOPS/DGS-NTA(Ni) = 50/40/10, 25 µM). Data were fitted as described in Methods using a two-state hindered anisotropy decay model for **a** and a hindered rotational diffusion model for (**b**) and (**c**). Insets show residuals of each fit.

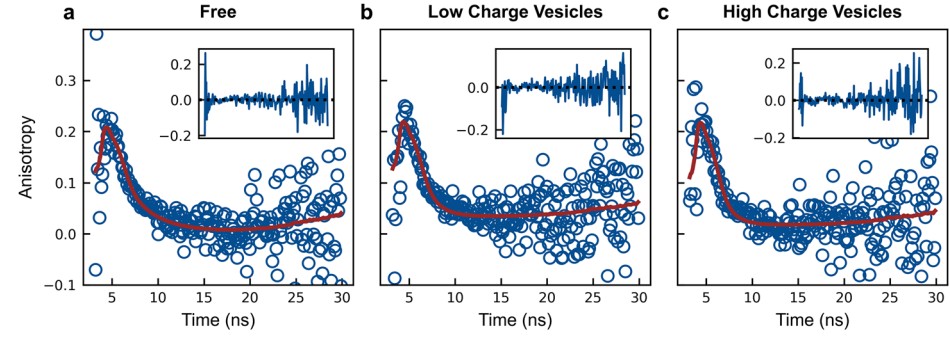

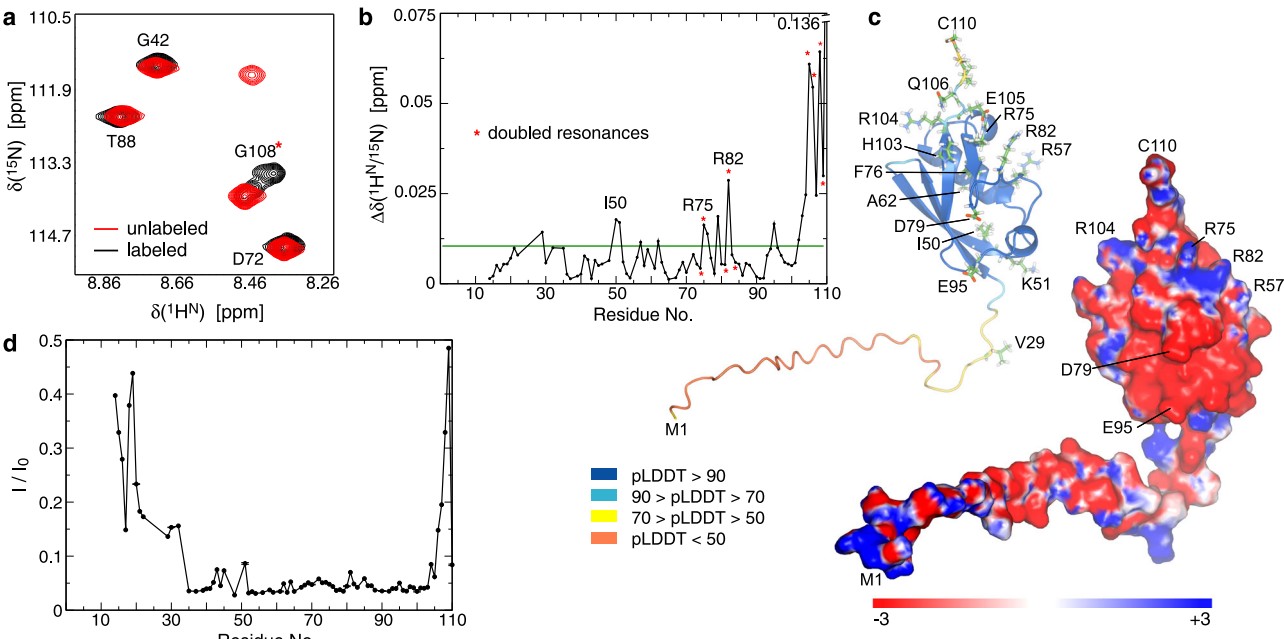

**Fig. 7 | Effect of fluorophore labeling and vesicle binding on hexahistidine model platform. a** Comparison of a representative region of the $^1H^N$-$^{15}N$ correlation spectra of fluorophore labeled and unlabeled platform (see also Figure S11). The resonance of G108 is split arising from the presence of residual unlabeled protein. **b** Combined $^1H^N$-$^{15}N$ chemical shift differences between the labeled and unlabeled platform. Green line indicates 0.01 ppm. **c** Residues with combined shift differences of >0.01 ppm are indicated in the AlphaFold3 model of the platform. The cartoon representation is color-coded according to the per-residue model confidence score (pLDDT). The surface representation is color-coded according to the electrostatic potential. The potential was calculated using APBS[56]. **d** Residual $^1H^N$-$^{15}N$ signal intensities of the fluorescently-tagged hexahistidine model platform in the presence of negatively charged vesicles (DOPC/DOPS/DGS-NTA(Ni) = 50/40/10, 25 mM). For the protein remaining free in solution, three distinct areas of signal reductions were observed, illustrating varying degrees of domain flexibility that are expected to be preserved in the vesicle-bound state. Vesicle-bound protein is tumbling too slowly to be observable.

This reduced two-state model (hindered rotational diffusion) provides a significantly improved fit to both the early and late anisotropy decay of the bound state (Fig. 6b,c, S10g–l), successfully capturing system behavior in the presence of both negatively charged and neutral vesicles. The primary distinction between the decay behavior of the two vesicle species lies in the relative contribution of individual model parameters ($r_\infty$ and $\theta_F$), which are modulated by the apparent presence or absence of membrane charge. It is important to note that although this model effectively describes the observed anisotropy decay, the physical interpretation of fluorophore motion is inherently model dependent. Alternative models may fit the data equally well, potentially yielding distinct but equally plausible mechanistic interpretations.

Taken together, the fluorescence decay data indicate the decrease in the steady-state fluorescence anisotropy occurs because of shift in the contribution weights from the individual components of the deconvolved motion. In the free state, there are significant contributions from both global and local motion of the platform construct and fluorophore, respectively. Upon binding, the overall decay is overwhelmed by the local motion of the fluorophore relative to the slow motion of the bound recombinant platform. This effect is present in the bound state of vesicles with both high and low negative membrane charge (Fig. 6a–c). A detectable decrease in steady-state anisotropy relative to the free state is only observed when the amount of negative membrane charge "visible" by the fluorophore exceeds a threshold, rather than resulting in a simple "cancellation" of the anisotropy increase induced by vesicle binding. While it is unclear if this phenomenon is directly responsible for the decrease in anisotropy of the original constructs, it suggests an overall molecular mechanism that locally restrains the fluorophore in the free state but is released in the presence of a negatively charged membrane. As a result, the steady-state anisotropy becomes lower than that of the unbound state; while coincidental in magnitude, this effect is highly robust and reproducible, enabling its use as a molecular reporter of binding.

To further analyze fluorophore mobility, we investigated the structural and dynamic properties of our platform by nuclear magnetic resonance (NMR) spectroscopy. First, we compared the NMR spectra of $^{13}C/^{15}N$-labeled versions of our His$_6$-SUMO-fluorophore construct with and without fluorophore to assess its impact on the SUMO domain. For most residues, NMR resonances remained virtually unchanged in the presence of fluorophore (Fig. 7a,b and Fig. S11). However, for a few residues, resonances shifted while often maintaining a minor resonance at the original peak position (Fig. 7a). Mass spectrometry showed a fluorophore labeling of ~78.6% (Fig. S13), suggesting that the minor peaks represent unlabeled protein. In AlphaFold3 models of our construct, the SUMO secondary structure ends at R104 (Fig. 1c) and the largest spectral differences are localized to E105-C110 in accordance with the C-terminal linkage of the fluorophore[40]. Within the SUMO domain, some chemical shift differences were detected (Fig. 7b) but they are small relative to well-defined SUMO-ligand interactions[41]. Mapping these chemical shift changes on the SUMO structure (Fig. 7c) produced no clear insight into specific contacts of the fluorophore with SUMO. With a net negative charge and conjugated ring system, the fluorophore likely makes differential contacts with the protein surface, but a clear preference was not discernible.

Next, we investigated NMR spectral changes of the His$_6$-SUMO-fluorophore construct in the presence of vesicles aiming to gain insight into the dynamic behavior of our platform. While the large particle size of the vesicle-bound platform construct did not allow the direct observation of the bound state by NMR, the protein fraction remaining free in solution experienced a relaxation contribution from its exchange with the vesicle-bound state. This contribution diminished the signal intensities of our construct to different degrees (Fig. 7d and Fig. S12), documenting a loose coupling between the N-terminal region (M1-E32), the SUMO domain, and the C-terminal region (Q106-C110). These transitions correlate with low confidence scores in the N- and C-terminal regions in AlphaFold3 models (Fig. 7c) and illustrate their dynamically unfolded nature. In the vesicle-

bound state, this loose coupling is expected to be preserved but an altered chemical and dynamic environment may change any fluorophore-SUMO contacts and dynamic fluorophore-SUMO coupling.

### Electrostatic interactions affect fluorophore mobility

The presence of a small positively charged patch on the surface of the SUMO folded domain (Fig. 7c) raised the question whether transient electrostatic interactions between this patch with the negatively charged fluorophore and/or membrane surface could contribute to the anisotropy decrease phenomenon. To address this, we expressed and purified two mutant His$_6$-SUMO-fluorophore constructs: a double mutant (R75A/R82A, dubbed 2RXA) and a quadruple mutant (R57A/R75A/R82A/R104A, dubbed 4RXA). The CD spectra of the mutated constructs showed no apparent changes in protein structure (Fig. S14). We measured the steady-state fluorescence anisotropy binding curves against increasing membrane DOPS content and environmental NaCl concentration, mirroring measurements from Fig. 5 (Fig. S15). Remarkably, both mutants continued to exhibit a decrease in anisotropy upon binding, while substantially expanding the sensitivity of the platform. Furthermore, the 4RXA variant exhibiting pronounced anisotropy decreases even with minimally charged membranes and under near-physiological salt conditions. These results indicate that the charged arginine patch may electrostatically restrict fluorophore mobility in the membrane-bound state, thereby limiting the extent of anisotropy reduction achievable under low-charge or high-salt conditions. However, the underlying mechanism remains uncertain, as the data do not distinguish whether this effect arises solely from fluorophore–protein electrostatics or from transient SUMO–membrane interactions that affect the overall binding conformation.

To further probe the steric conformation of membrane-bound His$_6$-SUMO-fluorophore constructs, we performed solution FRET measurements using vesicles doped with fluorescently labeled DOPE (headgroup-conjugated Rhodamine-B) as FRET acceptors for the OG488 protein label. Donor fluorescent decays were acquired as described in Methods under varying membrane DOPS content and NaCl concentration (Fig. S16). Mean fluorescence lifetimes were extracted from these decays and used to estimate FRET efficiencies[42]. We adopted this decay histogram-based analysis rather than direct fitting of the decay profiles because the vesicle-bound protein system produces highly complex FRET-induced multi-exponential decays, making it difficult to reliably resolve the underlying distribution of contributing decay components. The estimated FRET efficiency was therefore used as an inverse proxy for the mean donor–acceptor distance, and by extension, the average separation between the N-terminally conjugated donor fluorophore and the membrane surface. For all vesicle-containing samples, lipid-to-protein molar ratios were kept sufficiently high to ensure that the majority of protein was membrane-bound. In the absence of any vesicle interactions (such as for the hexahistidine-lacking "CONT" constructs), donor decays were well described by a single exponential with a mean lifetime of ~4 ns (consistent with literature for OG488[43]), whereas membrane binding resulted in highly heterogeneous, multi-exponential decays, reflecting the diverse conformational states adopted by the protein constructs in the membrane-bound state.

We observe a consistent increase in FRET efficiency for His$_6$-SUMO-fluorophore constructs bound to vesicles with increasing DOPS membrane content, a trend that is conserved across all mutant constructs tested and that tends to plateau at approximately 30–45% DOPS (Fig. S16k–m). This behavior indicates that increasing membrane charge promotes vesicle-bound conformational states in which the donor fluorophore resides closer to the membrane surface. However, because the FRET efficiencies saturate at values around ~0.3 (Fig. S16k–m), these conformations are unlikely to position the fluorophore directly at the membrane interface, which would be expected to yield substantially higher FRET efficiencies. Arginine-to-alanine substitutions within the SUMO domain preserve the overall dependence of FRET efficiency on membrane charge but progressively attenuate the magnitude of the increase, suggesting the conformational behavior of the fluorophore on the membrane surface is heavily dependent on electrostatics.

To further assess whether electrostatic interactions underlie these conformational changes, we performed analogous FRET experiments in which the membrane composition was held constant while the NaCl concentration was systematically varied (Fig. S17). Increasing the NaCl concentration from 0 to 100 mM resulted in a consistent decrease in FRET efficiency. At 300 mM NaCl, no detectable FRET signal was observed. The lack of FRET in this case likely results from a relaxed protein state, positioning the C-terminal fluorophore beyond the FRET radius, rather than a lack of binding (hexahistidine-tagged proteins are often captured on nickel columns at 500 mM NaCl). Consistent with this interpretation, progressive mutation of arginine residues (R57A/R75A and R57A/R75A/R82A/R104A) increasingly diminished FRET efficiencies across the same NaCl concentration range, indicating a more relaxed fluorophore state residing further from the membrane surface. Steady-state fluorescence anisotropy measurements corroborate this phenomenon, showcasing a distinct lower fluorescence anisotropy for mutant constructs regardless of NaCl concentration (Fig. S18a, b) in the free state. This lower fluorescence anisotropy suggests local enhanced mobility of the fluorophore that is less coupled to the mobility of the construct itself.

Taken together, these measurements suggest a composite mechanism where the anisotropy is modulated by both the global positioning of the protein relative to the membrane and the local steric freedom of the fluorophore. While salt concentration affects the average separation distance between the SUMO domain and the lipid bilayer by screening electrostatic interactions, the proximal arginine cluster also acts as an electrostatic tether that restricts the C-terminal conjugated fluorophore. Removal of this positive patch decouples the fluorophore from the protein core, resulting in enhanced local mobility, thereby potentially extending the utility of this platform to the near-physiological regime.

## Conclusions

Leveraging an unexpected decrease in fluorescence anisotropy upon binding offers a powerful approach to studying molecular interactions. A similar phenomenon has been previously reported in fluorescently labeled aptamers, where decreases in fluorescence anisotropy were utilized to quantify binding interactions across diverse systems[44–48]. In both cases, the decrease in anisotropy seemed to arise from an increased local flexibility, despite an overall increase in molecular size upon binding. This parallel suggests that the mechanisms governing anisotropy depolarization in aptamer systems may also apply to protein-lipid interactions and other contexts. By demonstrating that our recombinant platform can serve as a precise tool for measuring association with negatively charged membranes, this work greatly expands the scope of fluorescence anisotropy-based sensing, showcasing that a decrease in anisotropy can be a highly efficient biomolecular fluorescence probe. Furthermore, our findings show that there is no inherent need for lengthy structural investigations for an "ideal" rigid fluorescent labeling site when designing anisotropy probes. Instead, they demonstrate that previously ignored unstructured protein domains can be purposefully employed for binding measurements. Together, these findings reinforce the potential of fluorescence anisotropy as a precise and adaptable method for probing the governing parameters underlying biomolecular interactions.

While the low salt conditions and high membrane charge contents used in this study do not fully replicate the biological milieu, we believe that a controlled, high-throughput system can serve as an invaluable initial screening tool and streamline key parameters to be prioritized for rigorous testing in a more physiologically relevant setting. Although the requirement for charged lipid membranes somewhat limits our system, the intrinsically high negative surface charge of key organelles such as endosomes and mitochondria ensures that our conditions remain well-suited for capturing a broad range of biologically meaningful interactions. Furthermore, the enhanced anisotropy-decrease responses exhibited by the mutated versions of our model construct highlight opportunities to refine the platform and improve its relevance under near-physiological regime. Most importantly, the ability of our platform to detect AH sensitivity to certain lipid species in a

high throughput manner offers an invaluable new approach to uncovering the peptide code governing AH membrane binding. Discovering inducible amphipathic helix function and the parameters affecting its membrane association can allow us to distinguish between the protein domains responsible for binding from those involved in catalytic function, elucidating fundamental principles that govern membrane-associated biochemical pathways.

The overwhelming complexity of lipid species within organelle membranes is a fundamental hallmark of their biochemical identity, enabling a wide array of targeted protein-membrane interactions. Inducible amphipathic helices are key effectors of these interactions, initiating protein trafficking cascades by sensing specific lipid environments. Each residue of an amphipathic helix is likely solvated by a small number of lipids (on the order of one lipid per residue)[49], implying that a limited subset of lipids is necessary and sufficient for binding. Our high-throughput platform thus enables a powerful approach to investigate the residue- and peptide-level determinants necessary to detect such lipid environments, establishing a functional framework to probe how variations in individual lipid species can modulate peripheral membrane protein recruitment and define the peptide code underlying protein targeting, selectivity, and function.

## Materials and methods
### Materials
**For DNA cloning.** DH10B Competent cells were purchased from Thermo Scientific (Waltham, MA, USA). E.Z.N.A.® Plasmid DNA Maxi Kit was purchased from Omega Bio-tek (Norcross, GA, USA). Restriction digestion enzymes NdeI and NcoI-HF, Q5® High-Fidelity DNA Polymerase, NEBuilder® HiFi DNA Assembly Master Mix and Monarch® Spin DNA Gel Extraction Kit were purchased from New England Biolabs (Ipswich, MA, USA). Agarose SFR™ was purchased from VWR (Radnor, PA, USA).

**For protein expression, purification, and labeling.** BL21(DE3) Competent E. coli Cells were purchased from New England Biolabs. Sodium chloride (CAS #7647-14-5), sodium phosphate (CAS #7558-80-7), imidazole (CAS #288-32-4), 4-(2-hydroxyethyl)-1-piperazineethanesulphonic acid (HEPES) (CAS #75277-39-3), isopropyl-β-D-thiogalactopyranoside (IPTG) (CAS #367-93-1), tris(2-carboxyethyl) phosphine hydrochloride (TCEP-HCl) (CAS #51805-45-9) were purchased from VWR International. LB Miller broth and LB Miller agar were purchased from IBI Scientific (Dubuque, IA, USA). PureCube 100 Ni-NTA Agarose beads were purchased from Cube Biotech (Monheim, Germany). MilliporeSigma™ Amicon™ centrifugal filter units with 3 kDa MW cutoff were purchased from Sigma-Aldrich (St. Louis, MO, USA). Oregon Green™ 488 Maleimide, Alexa Fluor™ 647 $C_2$ Maleimide and dimethyl sulfoxide (DMSO, CAS #67-68-5) were purchased from Thermo Fisher Scientific. Isotopically enriched D-Glucose-$^{13}C_6$ (CAS #110187-42-3) and ammonium-$^{15}N$ chloride (CAS #39466-62-1) were purchased from Sigma-Aldrich. All reagents were used without additional purification.

**For liposomes.** 1,2-dioleoyl-*sn*-glycero-3-phosphocholine (DOPC), 1,2-dioleoyl-*sn*-glycero-3-phospho-L-serine (DOPS), 1,2-dioleoyl-*sn*-glycero-3-phosphoethanolamine (DOPE), 2-dipalmitoyl-*sn*-glycero-3-[(N-(5-amino-1-carboxypentyl)iminodiacetic acid)succinyl] (DGS-NTA(Ni)), 1,2-dioleoyl-*sn*-glycero-3-phosphoethanolamine-N-(lissamine rhodamine B sulfonyl) (18:1 Liss Rhod PE), 1′,3′-bis[1,2-dioleoyl-*sn*-glycero-3-phospho]-glycerol (Cardiolipin) and cholesterol (plant-derived) were purchased from Avanti Polar Lipids, Inc (Alabaster, AL, USA). Chloroform (CAS #67-66-3) was purchased from VWR International.

**For experimental measurements.** Corning® 96-well Half Area Black Flat Bottom Polystyrene NBS Microplate was purchased from Corning (Corning, NY, USA). Quartz fluorometer cell cuvettes were purchased from Starna Cells (Atascadero, CA, USA). Colloidal silica (CAS #112926-00-8) was purchased from Electron Microscopy Sciences (Hatfield, PA, USA).

### Methods
**Plasmids.** A master plasmid was designed and produced using a pET29b(+) backbone with an inserted domain encoding our platform (Twist Bioscience, San Francisco, CA, USA). The domain encodes a NdeI enzyme cleavage site, followed by a filler non-encoding region, a NcoI enzyme cleavage site, the sequence encoding for SUMO(21-98) (Uniprot: Q12306) with a mutated cysteine in the C-terminus, a sequence for the TEV cleavage site (ENLYFQS) and a hexa-histidine tag followed by a stop codon. For each peptide of interest, a separate plasmid was cloned by purchasing a linear double-stranded DNA fragment from Azenta (South Plainfield, NJ, USA) encoding for the corresponding peptide sequence, followed by the sequence for a GSGS soluble linker region and the SUMO(1-20) domain. The linear DNA fragment was cloned into the master plasmid using NEBuilder® HiFi DNA Assembly. After cloning, plasmids were transformed into DH10B chemically competent cells. DNA was isolated using the E.Z.N.A.® Plasmid DNA Maxi Kit and stored frozen in −20 °C. Sequences were confirmed using full plasmid sequencing by Plasmidsaurus (San Francisco, CA, USA). The pNatB (pACYCduet-naa20-naa25) plasmid (Plasmid #53613) and the MBP-superTEV (Plasmid #171782) encoding plasmid were acquired from Addgene. The master cloning plasmid [pET29b(+)_NdeI-NcoI_-SUMO(21-98)_CV] and the plasmid encoding the control construct [pET29b(+)_SUMO(1-98)_sTEV_His6] will be available on Addgene.

**Protein expression and purification.** Plasmids were transformed into BL21(DE3)-pNatB[25] chemically competent cells using standard transformation protocols. A saturated overnight starter culture from a single colony was used to inoculate 1 L of LB, which was grown at 37 °C to an OD of 0.5 and induced with 1 mM of IPTG for 4 h. Liquid colonies were centrifuged at 4000 g for 20 min at 4 °C and the pellets were resuspended in Ni-Reaction Buffer: 50 mM sodium phosphate at pH 8, 500 mM NaCl, 10 mM imidazole and sonicated for 10 min (2 s on, 2 s off, 50% amplitude) with a Qsonica Q125 (Qsonica, Newtown, CT, USA). Bacterial lysate was separated by centrifugation at 25,000 g for 30 min at 4 °C. Proteins were then purified by a 40-min rotating incubation with NTA-Ni beads. Bound proteins were eluted with 50 mM sodium phosphate, 500 mM NaCl, 300 mM imidazole (pH 8) and then buffer exchanged into 50 mM Tris, 0.5 mM EDTA, 0.5 mM TCEP (pH 8) using an Amicon centrifugal concentrator filter (3 kDa MWCO). TEV protease (Addgene plasmid #171782) was then added and incubated at 37 °C (not shaken) for 1 h and then overnight at 4 °C. The sample was buffer exchanged into Ni-Reaction buffer using an Amicon filter and incubated with NTA-Ni beads. This time the flowthrough was collected since the protein of interest has had its hexa-histidine tag cleaved. The sample was then further purified and buffer exchanged into 20 mM Hepes, 150 mM NaCl, 0.5 mM TCEP (pH 7.2) by size exclusion chromatography using a Cytiva Superdex 75 Increase 10/300 column with an AKTA Pure system (Cytiva, Marlborough, MA, USA). Peak fractions were collected, mixed, aliquoted, flash-frozen and stored in −80 °C.

**Protein labeling.** Oregon Green 488-maleimide and Alexa Fluor 647-maleimide dyes were dissolved in DMSO and stored in −20 °C. Labeling of the cysteine residue of each protein construct was performed at 100 μM protein concentration with a 5× molar excess of dye in 20 mM Hepes at pH 7.2, 150 mM NaCl, 0.5 mM TCEP (pH 7.2) overnight at 4 °C. The amount of DMSO in the reaction never exceeded 5%. Unconjugated dye was removed using a Cytiva Superdex 75 Increase 10/300 column with an AKTA pure system. Labeling ratios were measured using UV-vis spectroscopy with a Nanodrop One and ranged from 0.75 to greater than 0.95. Labeled proteins were flash frozen and stored at −80 °C.

**Protein expression $^{13}C/^{15}N$-labeled protein.** Plasmids were transformed into BL21(DE3)-pNatB[25] chemically competent cells using standard transformation protocols. A saturated overnight starter culture from a single colony was used to inoculate 1 L of LB, which was grown at 37 °C to an OD of 0.7. The culture was then spun down at 3000 g for 15 min in sterile bottles, and the resulting pellet was resuspended in 1 L of minimal media supplemented with 2 g of $^{13}C$ Glucose and $^{15}NH_4Cl$. The resuspended culture was grown at 37 °C for 1 h and then induced with 1 mM of IPTG for 4 h. The purification process was the same as described for unlabeled protein.

**Preparation of vesicles.** Lipid mixtures were prepared by combining the appropriate molar concentrations of lipid aliquots suspended in chloroform stored at −80 °C. The lipid mixture was then dried using a ThermoScientific Speedvac SPD130DLX evaporator with a Savant RVT5105 refrigerated vapor trap by centrifuging under vacuum for 3 h. The dried lipid mixture was dried under vacuum overnight and rehydrated the next day using the appropriate buffer and shaken at 37 °C for 45 min. The rehydrated mixture was then freeze-thawed five times, alternating between a dry-ice ethanol bath and a 42 °C water bath. The solution was then extruded 20 times through a track-etched Whatman Nuclepore Track-etch membrane (Cytiva, Marlborough, MA, USA) of desired pore diameter (50, 80 or 100 nm depending on experiment) using a Lipex 10 mL Extruder (Evonik, Burnaby, BC, CANADA) lipid extruder. The final vesicle size was confirmed by dynamic light scattering. Size is reported as the Z-average. For cardiolipin-containing vesicles, measurements were acquired within 1 day. In all other cases, measurements were acquired at maximum within 7 days.

**Tryptophan fluorescence.** Tryptophan fluorescence spectra were measured on a Horiba Fluorolog QM Spectrophotometer (Kyoto, Kyoto, Japan). The excitation beam monochromators were set at 280 nm with 8 nm slit widths. The emission monochromators were set to 300–420 nm with 4 nm slit widths. For all tryptophan fluorescence experiments, each datapoint corresponds to a sample volume of 2.1 mL under buffer conditions of 20 mM HEPES (pH 7.0) at 25 °C. To vary the lipid/protein ratio, we alter the lipid concentration in the sample. For the comparison experiments between tryptophan fluorescence and fluorescence anisotropy the protein concentration used in the samples was 125 nM for the Amphiphysin, CHMP4B and Huntingtin protein platforms and 100 nM for the Endophilin-B1 platform. To account for the increasing amount of scattering introduced by high lipid concentrations, we acquire a background fluorescence signal from the lipid sample prior to the addition of protein. After the addition of protein, the tryptophan fluorescence spectrum was acquired and then $n = 4$, 100 μL mixture samples were removed from the cuvette and distributed into 4 wells on a 96-well half-area plate. The fluorescence anisotropy values of these samples was immediately measured using the plate reader, with their averages and standard deviations shown in Fig. S2 and are used to generate the bound ratio values in Fig. 3. Spectra were fit to a linear combination of the fully bound and unbound spectra[32,50,51]:

$$S = bS_{bound} + (1 - b)S_{free}$$

Where $b$ is the molar fraction of the bound protein. The free-state spectrum is acquired with the protein solely suspended in buffer. We define the fully bound state as the condition in which further increases in lipid concentration no longer alter the observed spectra. Using a minimization algorithm, we determine the $b$ value for every point in between. The $b$ values are then used in nonlinear least-squares analysis to extract the binding kinetics parameters.

**Fluorescence anisotropy.** Fluorescence anisotropy measurements were performed on a black 96-well flat bottom half area plate (Corning) using a Biotek Synergy Neo2 plate reader (Agilent, Santa Clara, CA, USA) with

an 485/530 polarizing cube for protein samples labeled with Oregon Green 488 (all measurements except Fig. S6b) and a 620/680 polarizing cube for protein samples labeled with Alexa Fluor 647 (Fig. S6b). 100 μL of sample was added to each well, with a final protein concentration between 100 and 250 nM depending on the experiment (Fig. 1b: 125 nM, Fig. 2: 250 nM, Fig. 3: same as in Tryptophan Fluorescence section, Fig. 5: 250 nM). For all experiments, the buffering conditions were 20 mM HEPES (pH 7.0). NaCl was included only where indicated (Fig. 5, Fig. S15, Fig. S17). Measurements using the plate reader yielded emission intensities $I_{VV}$, $I_{VH}$ in the parallel and perpendicular polarization directions relatively to the excitation polarization, respectively. The anisotropy of a sample is then calculated as:

$$r = \frac{I_{VV} - GI_{VH}}{I_{VV} + 2GI_{VH}}$$

For each set of measurements where results are compared, the gain is kept constant. To calculate the G factor, we measure the anisotropy of fluorescein sodium salt at 1 nM and adjust G accordingly for the FA to be equal to the known value of 0.027[26]. For all fluorescence anisotropy data shown in this paper, values shown are averages from replicate wells, mixed independently using the same stocks of protein and lipids. Exception is the fluorescence anisotropy data in Fig. 3, where the average shown is described in the tryptophan fluorescence section. To extract the bound ratio of each datapoint we use the same method as with the tryptophan fluorescence data.

**Circular dichroism spectroscopy.** Circular dichroism measurements were collected on a JASCO J-1100 instrument with a 1 mm path length quartz cuvette (JASCO, Oklahoma City, OK, USA). Scans were taken from 195 to 300 nm at a 0.5 nm data pitch with 100 nm/min scan speed. The data integration time was 2 s, and each scan was performed twice. The buffer used for each experiment was 20 mM HEPES (pH 7.0) at 25 °C. Samples of vesicles and protein were allowed to equilibrate for at least 15 min before spectra were acquired. Each sample contained 8 μM of protein and was blanked by a buffer solution containing vesicles at the appropriate lipid concentration.

**Tethered vesicle assay.** Glass coverslips were cleaned with a modified RCA protocol and passivated with PEG-silane, as described previously with slight modification[52,53]. Briefly, glass slides were soaked in a concentrated KOH/peroxide bath, followed by a concentrated HCl/peroxide bath, each at 80 °C for 10 min. Once cleaned, the slides were passivated by surface treatment with a 7.5 mg/mL PEG-silane solution in isopropanol. The solution was comprised of 5% biotin-PEG-silane (5000 MW) and 95% of mPEG-silane (5000 MW). All materials for creating the PEG-silane solution were handled in a $N_2$ environment. Immediately before passivating the slide, 1% v/v of glacial acetic acid was added to catalyze the reaction between silanes and hydroxyl groups on the cover slip. 50 μL of the solution was added to the top surface of the cleaned cover slips, spread out, then placed into an oven at ~ 70 °C for 30–60 min. The slides were then rinsed under DI water to remove excess PEG-silane and dried under $N_2$. These slides were stored dry under air for a maximum of 1 week.

Imaging wells were created by placing 0.8 mm thick silicone sheets (Grace Bio-Labs) with 5 mm holes on top of the passivated slide. Each well was hydrated with 6 μg of NeutrAvidin (ThermoFisher Scientific, Waltham, MA, USA) in 30 μL of 20 mM HEPES (pH 7.0) and incubated for 10 min. After incubation, each well was rinsed to remove excess NeutrAvidin. Following this step, each well was rinsed with a solution containing vesicles to reach a 1 μM lipid concentration. After 10 min of incubation, excess vesicles were rinsed from the wells with the same buffer and the protein platform labeled with Oregon Green 488 was added at the appropriate concentration.

**Confocal microscopy.** Imaging was performed on a laser scanning confocal microscope (Leica Stellaris 5). Two excitation lasers were used:

488 nm and 638 nm for labeled protein and vesicles, respectively. The wavelength detection bandwidths used were 493–600 nm and 643–748 nm for labeled protein and vesicles, respectively. A Leica HC PL APO 63×, 1.4 NA oil-immersion objective was used to acquire images, and the zoom factor was set such that square pixel sizes of 70 nm were obtained. All images were acquired using a scan speed of 400 Hz.

**Fluorescence anisotropy decay.** Fluorescence anisotropy decay measurements were acquired with the DeltaDiode module of the Horiba Fluorolog-QM Spectrophotometer with a DeltaDiode DD-485L 479 nm pulsed excitation laser. Total sample size in all cases was 2.1 mL with buffering conditions of 20 mM HEPES (pH 7.0). Protein concentration was 25 nM, labeled with Oregon Green 488, while vesicle concentrations were 25 μM when present. Prior to each measurement, proper alignment of the excitation laser pulse and the emission polarizer was tested using an optically dilute colloidal silica solution. We considered the beams to be fully orthogonal if the polarization value was found to be 0.97 or greater. Parallel and perpendicular intensity emission responses were measured in sequence. Measuring time was set to be 5 min for each. After each measurement, we confirmed that the sample photobleaching was not significant (<2%). The emission spectrophotometer was set at 530 nm with 8 nm slit widths. Laser excitation pulse frequency was set at 500 kHz. Time binning was set at a window of 100 ns with 1024 total bins. Instrument response function for the parallel and magic polarizer angles were measured using an optically dilute colloidal silica solution for 1 min in each polarization direction. Analysis of the decays was performed in accordance with prior literature (Lakowicz, Principles of Fluorescence Spectroscopy, Chapter 11)[39]. Briefly, anisotropy decay models were each convolved with the corresponding instrument response functions and joint fitting of the parallel and perpendicular intensities was performed using a custom python code of nonlinear least square fitting (via the lmfit wrapper module of Python's SciPy library) using well-defined weights based on single photon counting statistics. Limiting values for each free parameter were set based on a physical understanding of the expected motion of the fluorophore in each case to avoid searching through the whole parameter phase space. For the double rotational correlation model with local hindered motion, the fluorescence lifetime ($\tau$) was limited to values between 3 and 5 ns, the fast rotational correlation time ($\theta_{\mathrm{F}}$) was limited between 0.1 and 2 ns, the slow rotational correlation time ($\theta_{\mathrm{P}}$) was limited between 2 and 10 ns for the case of the free protein and between 2 and 10,000 ns for the case of the bound protein, the time zero anisotropy ($r_0$) was limited between 0.1 and 0.4, the $\alpha$ parameter was limited between 0 and 1 and finally $I_0$ was set to be between $10^3$ and $10^6$. For the single hindered rotational correlation model, the fluorescent lifetime ($\tau$) was limited to values between 3 and 5 ns, the fast rotational correlation time ($\theta_{\mathrm{F}}$) was limited between 0.1 and 2 ns, the time zero anisotropy ($r_0$) was limited between 0 and 0.4, the long-time anisotropy $r_\infty$ was limited between 0 and 0.1 and finally $I_0$ was set to be between $10^3$ and $10^9$.

**FRET.** Vesicles for solution FRET experiments were made as described in the preparation of vesicles section, but with an added 2% molar weight of 18:1 Liss Rhod PE (Avanti A81150) as an acceptor and extruded through a 100 nm membrane. Donor lifetime decay measurements of the Oregon Green 488 protein fluorescent label were acquired with the DeltaDiode module of the Horiba Fluorolog-QM Spectrophotometer with a Delta-Diode DD-485L 479 nm pulsed excitation laser at 500 kHz. 10 nM of protein and 25 μM of vesicles were mixed for at least 5 min in a 1 cm quartz cuvette to a final volume of 2.5 mL at 25 °C. The emission spectrophotometer was set at 530 nm with 8 nm slit widths. Time binning was set at a window of 100 ns with 1024 total bins, and data was acquired until a maximum peak count of 10,000 photons was reached. The instrument response function was acquired using an optically dilute solution of colloidal silica. Due to the complexity of the decays of FRET exhibiting samples, mean fluorescence lifetimes of each sample were estimated

directly from the decay histogram as described by Fišerová and Kubala[42]. Briefly, the mean fluorescence lifetime $\langle\tau\rangle$ for each decay was calculated as $\langle\tau\rangle = \frac{\sum(N_i - noise)\cdot t_i}{\sum(N_i - noise)}$, where $N_i$ and $t_i$ are the photon count and time delay for the $i$th bin, respectively. For samples containing free protein, the fluorescence decay followed a clear single-exponential behavior and was reliably fit using a single-exponential decay model convolved with the instrument response function. In contrast, decays from membrane-bound samples exhibited highly complex lifetime distributions with substantial short-lifetime components, in some cases comparable to or shorter than the full width at half maximum of the instrument response function. These characteristics make deconvolution and model-based fitting extremely challenging and unreliable. Therefore, we analyzed these data using the mean fluorescence lifetime calculated directly from the decay histogram. This model-independent approach is well-suited for the analysis of complex fluorescence decays and large datasets. The resulting mean lifetimes were subsequently used to estimate FRET efficiencies as a general metric of donor–acceptor distance using $E = 1 - \frac{\langle\tau_{DA}\rangle}{\langle\tau_D\rangle}$, where $\langle\tau_{DA}\rangle$ and $\langle\tau_D\rangle$ are the mean fluorescence lifetimes of a sample mixed with acceptor possessing vesicles and free in solution, respectively.

**NMR experiments.** For backbone assignments, $^{13}$C/$^{15}$N-labeled protein samples with volumes of 320 μL were prepared at concentrations of 0.9 mM in 50 mM MOPS (pH 7.0), 3 mM TCEP and 6% D$_2$O solution (Fig. S11). In the presence of vesicles (Fig. S12), analogous samples were prepared containing 0.5 mM protein and 25 mM of vesicles with composition DOPC/DOPS/DGS-NTA(Ni): 50/40/10. Backbone assignments were achieved using HNCA, HNCO, HNCACB, CBCA(CO)NH and NOESY-HSQC (t$_m$=100 ms) experiments acquired on a Bruker Avance 700 (Bruker, Billerica, MA, USA) spectrometer at 35 °C. Data were processed and analyzed with the nmrPipe package and CARA[54]. The chemical shifts of untagged and tagged protein have been deposited to the BMRB database with accession codes 53091 and 53092, respectively. In the presence of vesicles, backbone assignments remained virtually unchanged (Fig. S12). Combined $^1$H$^N$-$^{15}$N chemical shift differences between untagged and tagged protein were calculated according to:

$$\Delta\delta(^1H^N/^{15}N) = \mathrm{SQRT}\left[(\Delta\delta(^1H^N)^2 + (\Delta\delta(^{15}N)/0.102)^2)/2\right]$$

**Mass spectrometry.** Matrix-Assisted Laser Desorption/Ionization Time-of-Flight (MALDI-TOF) intact protein analysis was performed at the USC Alfred E. Mann Multi-Omics Mass Spectrometry Core (MMSC; Director: Whitaker Cohn, Ph.D.). Protein samples were stored at −80 °C and then thawed on ice before analysis. Each sample was diluted 1:1 in a MALDI matrix solution composed of 15 mg/mL sinapinic acid, 70% acetonitrile, and 0.1% trifluoroacetic acid. A 2 μL aliquot of this mixture was spotted onto a Bruker MTP 384 Target Plate Ground Steel, and the samples were dried under nitrogen gas to promote evaporation and crystallization. The analysis was conducted using the Rapiflex Tissue-Typer MALDI-TOF mass spectrometer (Bruker Daltonik, Bremen, Germany), equipped with a smartbeam laser (Nd355 nm) and operated in linear positive mode to detect [M + H] species. Two optimized acquisition methods were employed to identify proteins across distinct mass ranges: 5–20 kDa. Calibration was achieved using appropriate standards, including Insulin ([M + H]$^+$ 5734.52 Da), Ubiquitin I ([M + H]$^+$ 8565.76 Da), Cytochrome C ([M + H]$^+$ 12360.97 Da, [M + 2H]$^{2+}$ 6180.99 Da), Myoglobin ([M + H]$^+$ 16952.30 Da, [M + 2H]$^{2+}$ 8476.65 Da). Data processing was conducted using Flex-Analysis software (Bruker Daltonik, Bremen, Germany) to perform data smoothing and background subtraction, effectively reducing noise and enhancing peak clarity.

**Binding kinetics**. To analyze inducible amphipathic helix peptide binding with membranes we considered the interaction as a partitioning equilibrium between the water/buffer phase and the lipid bilayer phase as first introduced by White et al.[55] and as applied by Robustelli and Baumgart[13]. Briefly the partition coefficient $K_p$ is given by

$$K_p = \frac{\frac{[P]_L}{[L]+[P]_L}}{\frac{[P]_W}{[W]+[P]_W}} \approx \frac{\frac{[P]_L}{[L]}}{\frac{[P]_W}{[W]}}$$

Where $[L]$ is the lipid concentration, $[W]$ is the water concentration and $[P]_L$ and $[P]_W$ are the protein concentrations in the lipid bilayer and in the buffer solution respectively, with the total amount of protein $[P]_t = [P]_L + [P]_W$, which allowed us to find the fraction of the protein in the bilayer:

$$K_p = \frac{\frac{[P]_L}{[L]}}{\frac{[P]_W}{[W]}} \Rightarrow [P]_W[L]K_p = [P]_L[W] \Rightarrow ([P]_t - [P]_L)[L]K_p = [P]_L[W] \Rightarrow [P]_t[L]K_p$$

$$= [P]_L\big([W] + [L]K_p\big) \Rightarrow f_B = \frac{[P]_L}{[P]_t} = \frac{K_p[L]}{[W] + K_p[L]}$$

For the high analysis shown in Fig. 2, we treated the fluorescence anisotropy signal of an intermediate bound state as a linear combination of the FA of the fully bound ($S_B$) and free ($S_F$) states:

$$S(L) = bS_B + (1 - b)S_F$$

We acquired fluorescence anisotropy data ($S(L)$) for different $[L]$ values and determined the $K_p$ by fitting the following equation:

$$S(L) = \frac{K_p[L]}{[W] + K_p[L]}S_b + \left(1 - \frac{K_p[L]}{[W] + K_p[L]}\right)S_f$$

using two fitting parameters $K_p$ and $S_B$. The water concentration was set to be $[W] = 55.5$ M, with $S_f$ determined for $[L] = 0$.

For the comparison measurements between fluorescence anisotropy and tryptophan fluorescence shown in Fig. 3, we transformed data acquired with each technique into bound ratio data as described in the "Tryptophan Fluorescence" Methods section. Then, the bound-ratio data were fitted using the above equation for $S(L)$, setting $S_f = 0$ with two fitting parameters $K_p$ and $S_B$. Since we assume a single bound state and, in all cases, we have either reached or are very close to the plateau of the binding curve, the resulting $S_B$ is essentially a scaling parameter proportional to the difference in signal between the free and bound states. As such, we re-normalized the resulting fitted curve and bound ratio data to 1 by dividing by the fitted value of $S_B$, considering error propagation for the data point error bars.

To analyze the fluorescence anisotropy data from the His₆-SUMO-fluorophore construct shown in Fig. 5 and Fig. S15, we used the depletion model, treating the protein as a ligand binding to independent binding sites, that are not always greater than the number of ligands in the sample[32]. The bound ratio was then calculated as:

$$f_B = \frac{1}{2[P]}\left(b_{max}[L] + [P] + \frac{1}{K_a} - \sqrt{\left(\left(b_{max}[L] + [P] + \frac{1}{K_a}\right)^2 - 4b_{max}[L][P]\right)}\right)$$

To fit the data, we again treated the fluorescence anisotropy signal of an intermediate bound state as a linear combination of the FA of the fully bound ($S_B$) and free ($S_F$) states, as before.

The binding data were fitted using nonlinear least square analysis with two free fitting parameters $K_a$ and $b_{max}$, with $S_B$ and $S_F$ being variable yet highly restricted to the anisotropy of the apparent fully bound and fully free states respectively.

## Statistics and reproducibility

Details on statistics and sample sizes can be found in the figure captions and the corresponding "Methods" sections where applicable.

## Reporting summary

Further information on research design is available in the Nature Portfolio Reporting Summary linked to this article.

## Data availability

All processed data supporting the findings of this manuscript are included within the manuscript and supplementary information. Raw data for all main figures are included in the Supplementary Data file. Protein sequences for all protein constructs used along with corresponding Uniprot accession numbers can be found in the Supplementary Data file. The chemical shifts of untagged and tagged protein have been deposited to the BMRB database with accession codes 53091 and 53092, respectively.

## Code availability

All data within the manuscript were analyzed using publicly available Python libraries.

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

## Acknowledgements

We would like to thank David M. Jameson for insightful conversations. P.J.C. and A.M. are supported by a grant from the National Institutes of Health (R35GM150716). D.H.J. and W.Z. are supported by a grant from the National Institutes of Health (R35GM147333). T.S.U. is supported by a grant from the National Institutes of Health (R01AG072442). Special thanks to Nuria Melisa Morales Garcia and Miranta Kouvari for their artistic contributions to Fig. 1. Intact protein analysis was performed at the USC Alfred E. Mann Multi-Omics Mass Spectrometry Core (Director: Whitaker Cohn, Ph.D.).

## Author contributions

A.M., M.Q., D.J., W.Z., T.U. and P.C. designed experiments. A.M., M.Q. and D.J. performed experiments. All authors consulted together on the interpretation of results and preparation of the manuscript.

## Competing interests

The authors declare no competing interest.
