## [Transparent Peer Review file · Communications Chemistry]

Recombinant protein platform for high-throughput investigation of peptide-liposome interactions via fluorescence anisotropy depolarization

Corresponding Author: Professor Peter Chung

Version 0:

Reviewer comments:

Reviewer #1

(Remarks to the Author)

A very interesting report communicating a paradoxical decrease in fluorescence anisotropy upon interaction with membrane vesicles for a chimeric protein construct consisting of a target peptide, SUMO, and a C-terminal tail labeled with a fluorescent probe via a cysteine-maleimide reaction.

The authors posit that the unexpected decrease in fluorescence anisotropy is due to an unknown conformational change that releases the fluorophore from a bound state. Figure 5 optimistically posits that this bound state involves the flexible linker region, which is then released, along with the fluorophore, due to interaction between the N-terminal target peptide and the vesicle. This is unlikely and extremely wishful thinking. The NMR data presented in Figure 5D indicates that the linker region and C-terminal tail are highly mobile - no conclusions beyond this can be drawn about structural and dynamic changes that occur upon vesicle binding. Instead, the mechanism by which the fluorescence anisotropy is decreased is obvious based on the NMR and fluorescence data as I will outline below.

First of all, Figure S13 shows that when the protein construct is fluorescently labeled, this does not just impact NMR signals in the C-terminal tail but throughout the SUMO protein as well, in particular a positively charged surface including R75 and R82 which the authors highlight. The negatively charged fluorophore binds through electrostatic interactions. This is not a non-specific interaction as the authors assert, because non-specific interactions do not give rise to specific chemical shift perturbations. This electrostatic interaction with SUMO is what increases fluorescence anisotropy, and it decreases when this interaction is broken.

The authors point out that the fluorescence anisotropy effect is highly salt-sensitive, being wiped out completely by a mere 100 mM NaCl, calling into question the range of physiologic conditions this assay can be employed (Figure 4c).

The fluorophore is released when the SUMO protein binds with negatively charged lipids like phosphatidylserine, which means that the vesicle must contain some anionic lipids for the assay to work (Figure 4). This calls into question the ability of the assay to assess lipid composition for peptide binding, since the addition of negatively charged lipids such as cardiolipin can also cause the assay to deviate significantly from results obtained alternatively by examining tryptophan fluorescence (Figure 3).

For the assay to work at all, it seems that binding of the peptide to the vesicles is needed, otherwise there would be not concordance with the tryptophan fluorescence assay (Figure 3). There is an appropriate negative control to prove this point (Figure S4). However, given that the mechanism by which the fluorescence anisotropy assay works is a secondary and additional interaction between SUMO and negatively charged vesicles, this phenomenon needs to be more accurately spelled out, along with its implications for interpreting peptide binding and dependence on lipid composition.

As such, Figure 1 is highly misleading.

Until the above points are addressed, I don't think that the study can be published. Otherwise, the study is expertly done and of interest, so it could potentially be published, but the limitations of the assay must be clearly pointed out.

Reviewer #2

(Remarks to the Author)

The manuscript "Recombinant protein platform for high-throughput investigation of peptidoliposome interactions via fluorescence anisotropy depolarization" describes a high throughput approach to measure binding affinities of amphipathic helices to lipid bilayer membranes.

For that purpose, the authors recombinantly attach to those helices the SUMO protein, which helps expression and solubility, as well as a synthetic fluorophore C-terminal to the SUMO protein. Surprisingly, the authors find that upon membrane binding, the fluorescence anisotropy of the fluorophore decreases when the amphipathic helix binds to the membrane.

The authors investigate the role of ionic strength, vesicles charge, DOPE content, and vesicle size and are able to recapitulate known effects of these determinants in contributing to membrane binding affinity. The authors further characterize the role of cholesterol and find a surprising increase of binding for the amphiphysin H0 helix for increasing cholesterol content.

This study furthermore systematically evaluates the performance of the binding sensor, using tryptophan fluorescence as a complementary approach. Finally, they investigate the mechanisms behind the binding sensor behavior and find that the signal is based on a repulsive interaction between the fluorophore and the membrane surface, leading to increased rotational mobility after membrane binding.

Overall, the study is carefully conducted and provides an interesting new method for the evaluation of membrane binding behavior of amphipathic helices. A large focus of this manuscript is the development and characterization of the method, whereas scientific insight into the biophysical chemistry of amphipathic helices interacting with membranes is more incremental. While a significant advantage of this new technique is the high throughput nature, a significant disadvantage is that high negative membrane charge and low ionic strength are required, possibly excluding use under physiologically relevant conditions. Nevertheless, the technique presents a valuable and creative new approach.

I have the following additional comments for the authors to consider:

- 1) The authors conclude that long range repulsive electrostatic interactions can significantly affect the fluorescence anisotropy of the conjugated fluorophore. Nevertheless, there does not seem to be concern that this interaction affects the membrane binding affinity of the amphipathic helix. Would it be possible to discuss this via free energies of binding relative to the fact that the electrostatic interaction with the fluorophore must be on the order of kT to affect the anisotropy?
- 2) I find the breakdown between five main figures and several supplementary figures distracting. I recommend to move the supplementary figures into the main text.
- 3) The authors refer to their method and / or experimental system as a 'platform' but it is not always clear what is meant. Nor is the use of the word in this sense always consistent: for example, on page 4 they refer to an increased rotational diffusion of the platform upon membrane binding, when it is just the synthetic fluorophore the rotational diffusion of which is being probed. Confusingly, the SUMO protein without a conjugated peptide is also referred to as a platform. Extending this problem, later on, the AH-SUMO conjugation without the fluorophore is also referred to as 'platform'. I recommend replacing the word 'platform' with something more specific – perhaps an acronym for the AH-SUMO-fluorophore molecule and to be more accurate whenever reference is made to this molecule or the method.
- 4) While the authors claim that peptides where phenylalanine has been replaced with a tryptophane show the same membrane binding behavior as detected by their anisotropy-based method, it is to be expected that a tryptophane containing peptide binds more strongly to the bilayer than a phenylalanine containing peptide (see e.g. Whimley White hydrophobicity scale). This should be discussed.
- 5) The sentence "Taken together, the fluorescence decay data indicate the decrease in steady-state fluorescence anisotropy occurs because there is a transition" sounds a bit circular – perhaps it would help if the authors explained at this point what they mean by 'transition'.
- 6) Next, I'm not sure what is meant by 'meaningful contribution'. Would it be more accurate to replace 'meaningful' by 'measurable', 'detectable', or 'significant'?
- 7) The fragment: "the overall decay is dominated by the local motion of the fluorophore and there is a concomitant increase in local mobility" sounds like repeating the same thing twice (local motion and local mobility increase).
- 8) In the following sentence the authors then refer to "oscillatory motion" of the fluorophore. This does not seem to be physically correct as oscillatory motion is deterministic, whereas the authors are observing Brownian motion – a stochastic process.
- 9) The value of the NMR experiment is questionable as membrane binding could not be detected with this approach. Perhaps it would help to motivate this experiment more clearly.
- 10) The authors write that the technique is not limited by an upper anisotropy. This is a bit misleading in two ways. First, any fluorescence anisotropy measurement is bound by a fundamental upper limit. What the authors mean to express is that their anisotropy decreases upon binding – but there is of course a lower limit to fluorescence anisotropies as well. Perhaps the

authors could reword this section to explain more clearly what they have in mind.

11) I'm not sure it is accurate to say that the high throughput nature of the technique 'outweighs' the technical limitation that high membrane charge and low ionic strength is required. Instead these are simply advantages and disadvantages of the technique that future users need to be aware of.

Version 1:

Reviewer comments:

Reviewer #1

(Remarks to the Author)

In the revised manuscript, the authors are more transparent in acknowledging the requirement for low salt concentration and negatively charged lipids in their experimental platform. However, this phenomenon is not mentioned in the abstract, and the authors still try to minimize its significance, postulating in their rebuttal to me: "there is likely a secondary mechanism by which the

decreases in anisotropy are observed upon binding that are not dependent on the charge of the vesicle surface. This observation was key in proposing a potential steric hindrance between the N-terminus of the protein and the fluorophore."

The above statement by the authors is faulty because all of the decreases in anisotropy that are observed upon vesicle binding do in fact depend on the presence of negative charges on the vesicle surface. The authors cite Figures 5b, 7b and S6a as their counter evidence. (Actually, Figure S6a is not evidence of this phenomenon because binding in this figure occurs via His-tag-nickel ion interaction. I think the authors mean Figure S6b.) In all of these figures, they wrongly assert that the lipid mixture of DOPC/DGS-NTA(Ni) is neutral. However, this is incorrect. While NTA has a net charge of -2 from an amine group and three carboxylates that appear to cancel out the positive charge from Ni²⁺, binding of Ni²⁺ actually displaces a proton from the amine group, so DGS-NTA(Ni) has in fact a -1 charge. Thus, the authors mistakenly believe that these observations challenge the need for an electrostatic interaction to observe fluorescence anisotropy changes; however, this is not the case.

The "steric hindrance" mechanism between the N-terminal tail of the protein and the fluorophore arises from a misinterpretation of the NMR data presented in Figure 8, a point I made in my initial review. A cursory glance at the structures shown in Figures 7 and 8 show that this is impossible, with the N-terminal tail and the fluorophore-labeled C-terminal tail pointing away from opposite ends of the SUMO domain. The model structure presented in Figure 7d is thus impossible. The NMR data in Figure 8 is misinterpreted by supposing that because the N- and C-terminal tails show similar reduction in intensity induced by vesicle binding, they must be somehow close to each other or interacting. However, this is entirely not the case, all you can conclude is that the N-terminal tail is flexible and so is the C-terminal tail. In fact, this phenomenon is better explained by the SUMO domain itself forming transient electrostatic interactions with the negatively charged vesicles, with increasing mobility observed in the tails as one moves further away from the SUMO domain. In contrast, if only the N-terminal tail were tethered to the vesicles as suggested by the authors in Figure 7, one would not expect to see such restriction in the proximal C-terminal tail.

Thus, the structural model proposed by the authors is further discredited, the structural models presented in the figures are misleading, and the importance of electrostatic interactions is downplayed. I still stand by my structural model as the only reasonable explanation thus far. The authors also wrongly interpret the chemical shift perturbations seen upon attachment of the fluorophore to the C-terminus to exclude a specific interaction between fluorophore and SUMO domain. Perturbations typically occur throughout the protein domain, and the largest perturbations do not always map to the binding site. Side chain chemical shift perturbations are better at delineating binding sites than backbone chemical shifts, but this data is not presented.

In conclusion, the observed fluorescence anisotropy change is very dependent on electrostatic forces that initially restrict the motion of the fluorophore via interaction of the negatively charged fluorophore-C-terminal tail with a limited positively charged patch on the SUMO domain (one can see it is limited from the authors' new AlphaFold3 model). The fluorophore is released by electrostatic interaction between the SUMO domain and negatively charged vesicles. However, this electrostatic interaction is not strong enough on its own and requires additional binding of N-terminal vesicle-binding amphipathic helices that the authors are proposing to study.

The authors have thus not refuted my structural model or supported their own, and as such I still cannot recommend this manuscript for publication in its present form.

Reviewer #2

(Remarks to the Author)

The authors have thoughtfully and successfully addressed my comments and concerns and have significantly revised their manuscript text.

I just have one more comment: it seems that the other referee has suggested an intriguing, specific, alternative mechanism. It

may be worthwhile for the authors to (re-consider) the possibility of being more specific, when they suggest - in the revised manuscript text - that alternative mechanisms are possible.

Version 2:

Reviewer comments:

Reviewer #1

(Remarks to the Author)

The authors have made extensive changes addressing all of my concerns and have added considerable additional experimental data. I now recommend for publication.

Reviewer #2

(Remarks to the Author)

The authors have addressed all of my concerns and comments and have thoughtfully addressed and responded to the other referee's comments and significantly revised the manuscript.

Reviewer comments are in bold.

Author responses are in regular font.

Removed text from the original manuscript is in red.

All revisions in manuscript are in blue.

Reviewer #1

A very interesting report communicating a paradoxical decrease in fluorescence anisotropy upon interaction with membrane vesicles for a chimeric protein construct consisting of a target peptide, SUMO, and a C-terminal tail labeled with a fluorescent probe via a cysteine-maleimide reaction.

The authors posit that the unexpected decrease in fluorescence anisotropy is due to an unknown conformational change that releases the fluorophore from a bound state. Figure 5 optimistically posits that this bound state involves the flexible linker region, which is then released, along with the fluorophore, due to interaction between the N-terminal target peptide and the vesicle. This is unlikely and extremely wishful thinking. The NMR data presented in Figure 5D indicates that the linker region and C-terminal tail are highly mobile - no conclusions beyond this can be drawn about structural and dynamic changes that occur upon vesicle binding. Instead, the mechanism by which the fluorescence anisotropy is decreased is obvious based on the NMR and fluorescence data as I will outline below.

First of all, Figure S13 shows that when the protein construct is fluorescently labeled, this does not just impact NMR signals in the C-terminal tail but throughout the SUMO protein as well, in particular a positively charged surface including R75 and R82 which the authors highlight. The negatively charged fluorophore binds through electrostatic interactions. This is not a non-specific interaction as the authors assert, because non-specific interactions do not give rise to specific chemical shift perturbations. This electrostatic interaction with SUMO is what increases fluorescence anisotropy, and it decreases when this interaction is broken.

To better understand the significance of the fluorophore-induced chemical shift changes in the SUMO domain, we have mapped the shift changes on the AlphaFold3 model of our construct. As shown in Figure 8, there is no clear pattern recognizable that would indicate specific fluorophore-SUMO contacts. Nonetheless, our use of the term “non-specific” is perhaps too broad as the chemical shift changes are indeed observed for only a few residues. We have therefore deleted this term in the manuscript and updated the main text to reflect the fact that the fluorophore likely makes differential contacts with the protein surface, but with no clear preference discernible. To ensure cohesiveness in the narrative of the manuscript the following paragraph on page 12 was reworked:

~~To corroborate our mechanistic interpretation that fluorophore mobility increases in the bound state of the platform we investigated the structural and dynamic properties of our platform by nuclear magnetic resonance (NMR) spectroscopy. First, to verify that fluorophore interactions with the rest of the platform remained non-specific, we compared the NMR spectra of $^{13}\text{C}/^{15}\text{N}$ -labeled versions of our hexahistidine model platform with and without a fluorophore. Spectral differences predominantly localized to the C-terminal region where the fluorophore is conjugated (Figure S13), although some resonance signals within the SUMO domain were doubled indicating a non-negligible fluorophore proximity effect. Nevertheless, the small chemical shift differences between split resonances show that the structure of the folded SUMO domain remained virtually unchanged. Moreover, since specific SUMO-ligand interactions usually produce much larger shift changes³⁸ and resonance doubling was also observed near the C-terminal conjugation site, we deem any SUMO-fluorophore interaction to be unspecific.~~

To further analyze fluorophore mobility, we investigated the structural and dynamic properties of our platform by nuclear magnetic resonance (NMR) spectroscopy. First, we compared the NMR spectra of $^{13}\text{C}/^{15}\text{N}$ -labeled versions of our His₆-SUMO-fluorophore construct with and without fluorophore to assess its impact on the SUMO domain. For most residues, NMR resonances remained virtually unchanged in the presence of fluorophore (Figure 8a-b and Figure S10). However, for a few residues, resonances shifted while often maintaining a minor resonance at the original peak position (Figure 8a). Mass spectrometry showed a fluorophore labeling of ~78.6% (Figure S12), suggesting that the minor peaks represent unlabeled protein. In AlphaFold3 models of our construct, the SUMO secondary structure ends at R104 (Figure 1c) and the largest spectral differences localized to E105-C110 in accordance with the C-terminal linkage of the fluorophore⁴⁰. Within the SUMO domain, some chemical shift differences were detected (Figure 8b) but they are small relative to well-defined SUMO-ligand interactions⁴¹. Mapping these chemical shift changes on the SUMO structure (Figure 8c) produced no clear insight into specific contacts of the fluorophore with SUMO. With a net negative charge and conjugated ring system, the fluorophore likely makes differential contacts with the protein surface, but a clear preference was not discernible.

The fluorescence anisotropy decrease mechanism proposed by the reviewer is intriguing. However, this mechanism does not seem to be strongly supported by our data. Should the decrease in anisotropy be a byproduct of the collapse of an electrostatic interaction between the fluorophore and positively charged amino acids within the main body of the SUMO (induced by the negative charge of the membrane), then that interaction would remain intact when the construct is bound to neutral vesicles. To wit, we see no change in the steady state fluorescence anisotropy upon titration of more vesicles with little to no charge (DOPC/DGS-NTA(Ni) : 90/10, Figure 5b), yet we detect binding (Figure S6a). In contrast, model fits of our anisotropy decay and NMR data suggest that the diffusion of the globular domain of the SUMO protein is coupled to that of the vesicle (Figure 7b). Hence, there is likely a secondary mechanism by which the decreases in anisotropy are observed upon binding that are not dependent on the charge of the vesicle surface. This observation was key in proposing a potential steric hindrance between the N-terminus of the protein and the fluorophore. We readily admit that this interpretation is

dependent on the model by which we fit our data. As such, we will clearly delineate assumptions made that are in support of our hypothesis.

Below is added text in page 12 of the revised manuscript:

The primary distinction between the decay behavior of the two vesicle species lies in the relative contribution of individual model parameters (r_{∞} and θ_F), which are modulated by the apparent presence or absence of membrane charge. **It is important to note that although this model effectively describes the observed anisotropy decay, the physical interpretation of fluorophore motion is inherently model dependent. Alternative models may fit the data equally well, potentially yielding distinct but equally plausible mechanistic interpretations.**

The authors point out that the fluorescence anisotropy effect is highly salt-sensitive, being wiped out completely by a mere 100 mM NaCl, calling into question the range of physiological conditions this assay can be employed (Figure 4c).

The reviewer is correct to point out this significant limitation of our assay. We do not offer the technique as a rigorous tool by which the binding affinity of amphipathic helices can be measured in biological-like conditions; instead, we offer a high-throughput probe that is the entry point to a comprehensive and multi-step parameter search for each protein of interest. Low salt conditions have been used in the past by Robustelli and Baumgart¹ as a starting point for uncovering alpha-helical lipid specific binding preferences of the Endophilin Helix 0 isoforms. Our system offers a way to significantly shorten such undertakings which usually would take years.

We have altered the following text in page 14 of our manuscript to accurately reflect this limitation of our assay.

While the low salt conditions and high membrane charge concentrations used in this study do not fully replicate the biological milieu, we believe ~~the advantages of that~~ a controlled, high-throughput system ~~strongly outweigh these limitations.~~ **can serve as an invaluable initial screening tool and streamline key parameters to be prioritized for rigorous testing in a more physiologically relevant setting. While the requirement for charged lipid membranes somewhat limits our system, the intrinsically high negative surface charge of key organelles such as endosomes and mitochondria ensures that our conditions remain well-suited for capturing a broad range of biologically meaningful interactions.** Most importantly, the ability of our platform to detect AH sensitivity to certain lipid species in a high throughput manner offers an invaluable new approach to uncovering the peptide code governing AH membrane binding. Discovering inducible amphipathic helix function and the parameters affecting its membrane association can allow us to distinguish between the protein domains responsible for binding from those involved in catalytic function, elucidating fundamental principles that govern membrane-associated biochemical pathways.

The fluorophore is released when the SUMO protein binds with negatively charged lipids

like phosphatidylserine, which means that the vesicle must contain some anionic lipids for the assay to work (Figure 4). This calls into question the ability of the assay to assess lipid composition for peptide binding, since the addition of negatively charged lipids such as cardiolipin can also cause the assay to deviate significantly from results obtained alternatively by examining tryptophan fluorescence (Figure 3).

This is indeed another limitation of our assay and we are clear about that in this manuscript. Adding increasing amounts of negative charge in the membrane may impact peptide binding to a degree where the fluorescence anisotropy data only qualitatively agree with the tryptophan fluorescence curves such as in the case of Huntingtin with cardiolipin containing vesicles (Figure 3d). However, this effect seems to be peptide specific as the same does not happen with CHMP4B (Figure 3b), where the quantitative agreement between the two techniques is maintained even at high cardiolipin concentrations. Overall, our assay seems to maintain at minimum a qualitative agreement with the tryptophan fluorescence results across all parameters tested.

To further outline this limitation of our assay the following text has been added in page 10 of our revised manuscript:

To further test this idea, we introduced a charge-altering mutation into the fluorescently-tagged C-terminal domain (R104H). This mutation narrowed the dynamic detection window without affecting the overall binding isotherm remained (Figure 6). **Overall, the presence of some membrane charge detectable by the fluorophore is crucial for measurement via anisotropy decrease (Figure 5c). Increasing NaCl concentration in the experimental buffer sharply increases the membrane charge requirement, with concentrations over 100 mM abolishing any measurement regardless of membrane charge.** These findings suggest that the decrease in fluorescence anisotropy is directly linked to membrane charge exposure and indicates that local conformational changes within the fluorescently-tagged C-terminal domain can modulate this effect without affecting overall binding.

For the assay to work at all, it seems that binding of the peptide to the vesicles is needed, otherwise there would be not concordance with the tryptophan fluorescence assay (Figure 3). There is an appropriate negative control to prove this point (Figure S4). However, given that the mechanism by which the fluorescence anisotropy assay works is a secondary and additional interaction between SUMO and negatively charged vesicles, this phenomenon needs to be more accurately spelt out, along with its implications for interpreting peptide binding and dependence on lipid composition.

We agree that the mechanism is important for defining the conditions in which this assay is accessible. We have attempted to do so with the techniques available to us and tried to define the lipid ranges / salt conditions in which this assay will work. While the mechanism is the result of our best interpretation, we readily accept that the actual underlying phenomenon governing the observed anisotropy transitions might be different than our model. More importantly, we have taken steps to clearly define in the text the salt conditions and lipid composition ranges in which this tool will work, and we would advise any reader to keep within those defined

conditions for our interpretations to apply. We attempted to clearly express this sentiment in pages 10 and 14 of the revised manuscript as shown in the responses to the two previous comments.

As such, Figure 1 is highly misleading.

The reviewer correctly points out an oversight in the figure description on our part. The intended purpose of Figure 1a is to display the individual engineered components that make up our recombinant construct. We did not mean to mislead that the arrangement shown here has anything to do with a realistic state in solution. To correct this, we have changed the caption to the following.

Figure 1 | Our recombinant protein platform exhibits a decrease in fluorescence anisotropy upon binding to a vesicle (a) Schematic representation of the SUMO protein platform. Peptides of interest (dark blue) are engineered at the N-terminus, linked to SUMO via soluble linker (light blue) and a mutated cysteine to facilitate fluorophore attachment (green). A cleavable C-terminal hexahistidine tag (not depicted) facilitates purification and cleaved prior to usage. **Schematic represents individual engineered protein domains, not a conformation of the construct's free state.** (b) An unexpected decrease in fluorescence anisotropy is observed when measuring the N-terminal domain of Amphiphysin engineering to our platform upon increasing titration of 67 nm diameter DOPC/DOPS (70/30) vesicles. Data was fit using a membrane partitioning model (blue) as described in Methods.

Reviewer #2

The manuscript “Recombinant protein platform for high-throughput investigation of peptide liposome interactions via fluorescence anisotropy depolarization” describes a high throughput approach to measure binding affinities of amphipathic helices to lipid bilayer membranes.

For that purpose, the authors recombinantly attach to those helices the SUMO protein, which helps expression and solubility, as well as a synthetic fluorophore C-terminal to the SUMO protein. Surprisingly, the authors find that upon membrane binding, the fluorescence anisotropy of the fluorophore decreases when the amphipathic helix binds to the membrane.

The authors investigate the role of ionic strength, vesicles charge, DOPE content, and vesicle size and are able to re-capitulate known effects of these determinants in contributing to membrane binding affinity. The authors further characterize the role of cholesterol and find a surprising increase of binding for the amphiphysin H0 helix for increasing cholesterol content.

This study furthermore systematically evaluates the performance of the binding sensor, using tryptophan fluorescence as a complementary approach. Finally, they investigate the mechanisms behind the binding sensor behavior and find that the signal is based on a repulsive interaction between the fluorophore and the membrane surface, leading to

increased rotational mobility after membrane binding.

Overall, the study is carefully conducted and provides an interesting new method for the evaluation of membrane binding behavior of amphipathic helices. A large focus of this manuscript is the development and characterization of the method, whereas scientific insight into the biophysical chemistry of amphipathic helices interacting with membranes is more incremental. While a significant advantage of this new technique is the high throughput nature, a significant disadvantage is that high negative membrane charge and low ionic strength are required, possibly excluding use under physiologically relevant conditions. Nevertheless, the technique presents a valuable and creative new approach.

I have the following additional comments for the authors to consider:

1) The authors conclude that long range repulsive electrostatic interactions can significantly affect the fluorescence anisotropy of the conjugated fluorophore. Nevertheless, there does not seem to be concern that this interaction affects the membrane binding affinity of the amphipathic helix. Would it be possible to discuss this via free energies of binding relative to the fact that the electrostatic interaction with the fluorophore must be on the order of kT to affect the anisotropy?

This is an intriguing point raised by the reviewer. Indeed, we have not considered the possibility that the fluorophore interaction leading to the decrease in anisotropy affecting peptide binding. While the phenomenon described by this idea is certainly valid and likely present in our overall system, we have a good reason to believe that its contribution is not significant enough to be detectable by current techniques. Specifically, for the Endophilin B1 binding to vesicles comprised of DOPC/DOPS : 70/30 extruded through 50 nm pore size membranes, we measured a partitioning coefficient of $k_p = (4 \pm 0.6) \times 10^5$ and $(3.5 \pm 1.1) \times 10^5$ tryptophan fluorescence and fluorescence anisotropy respectively. We can convert these partitioning coefficients into Gibbs free energy of binding using the formula $\Delta G = -RT \ln(k_p)$ which yields a $\Delta G = -7.64 \pm 0.09$ kcal/mol and $\Delta G = -7.6 \pm 0.2$ kcal/mol at $T=298$ K. Robustelli and Baumgart measure a $\Delta G \sim -6.9$ kcal/mol for the Endophilin B1 domain binding to vesicles comprised of DOPC/DOPS : 75/25 also extruded through 50 nm pore size membranes of $\Delta G \sim -6.9$ kcal/mol, a value closely resembling our measurement to vesicles of similar composition. The slightly lower value calculated by Robustelli and Baumgart is to be expected as in their paper show that the Endophilin B1 has a strong affinity for more electronegative membranes, and their vesicles contain 5% less DOPS than ours.

Unfortunately, theoretically estimating the free energy decrease of the fluorophore upon binding is not trivial. The overall motion of the fluorophore is accurately described by the two-state hindered anisotropy decay model in the free state, while in the bound state the model reduces to the hindered rotational diffusion model (Figure 7a-c). The complexity of the motion in the free state, as well as the nature of the conformational expansion experienced by the fluorophore upon binding, limit the avenues that can be explored in describing this theoretically via free energies. Even though we are not able to describe this effect in a theoretical manner, the experimental agreement of our measured free energies to those measured by Robustelli and Baumgart assures us that to at least the experimentally measurable extent, the electrostatic

effects leading to the increased fluorophore mobility upon binding do not significantly affect the binding of the inducible amphipathic helix.

2) I find the breakdown between five main figures and several supplementary figures distracting. I recommend to move the supplementary figures into the main text.

We tried to maintain a balance between what we thought was essential figures to the narrative of the paper, and the ones that have a more supporting nature. We see how the extensive length of the supplement might make it distract when following the main text. To address this, we moved 4, 5 and 10 into the main text. We think that this new arrangement allows for a more cohesive presentation of the results, and we thank the reviewer for pointing it out.

3) The authors refer to their method and / or experimental system as a ‘platform’ but it is not always clear what is meant. Nor is the use of the word in this sense always consistent: for example, on page 4 they refer to an increased rotational diffusion of the platform upon membrane binding, when it is just the synthetic fluorophore the rotational diffusion of which is being probed. Confusingly, the SUMO protein without a conjugated peptide is also referred to as a platform. Extending this problem, later on, the AH-SUMO conjugation without the fluorophore is also referred to as ‘platform’. I recommend replacing the word ‘platform’ with something more specific – perhaps an acronym for the AH-SUMO-fluorophore molecule and to be more accurate whenever reference is made to this molecule or the method.

We thank the reviewer for pointing out this discrepancy in our text. We agree that the term platform, as previously used, was too general and may have caused confusion. To clarify our meaning, we have replaced platform with more precise terms, such as protein construct or the specific name of the construct used, where appropriate. This change helps distinguish the general concept we intended to convey with platform: namely, the overarching protein construct featuring a replaceable N-terminal segment and a characteristic decrease in fluorescence anisotropy upon binding.

The changes are as follows in the revised manuscript:

Page 4:

We expected the apparent larger size of the bound ~~platform~~ ~~construct~~/vesicle system (upon inducible amphipathic helix binding) to lead to elevated fluorescence anisotropy

The observed anisotropy decrease indicates a counterintuitive increase in rotational diffusion of our recombinant ~~platform~~ ~~construct~~ when bound to the much larger vesicle.

This increase in local mobility dominates the overall rotational motion of the bound ~~platform~~ ~~construct~~/vesicle system

Page 5:

We expressed and purified our recombinant platform ~~s~~ constructs presenting peptides corresponding to the inducible amphipathic helices

Page 7:

“We also performed control measurements using a platform construct without conjugated peptides to confirm that the binding isotherms exclusively reflect interactions with the inducible amphipathic helices (Figure S4).”

Page 8:

We do not expect the binding isotherms between the two techniques to always be identical, as tryptophan fluorescence is a direct measure of binding while our platform's fluorescence anisotropy reflects the proximity of the platform construct to the membrane. Nevertheless, we confirmed binding via the formation of an amphipathic helix (for all peptides) using circular dichroism spectroscopy (Figure S6).”

To achieve this, we replaced the peptide corresponding to an inducible amphipathic helix with hexahistidine (His₆-SUMO-fluorophore).

Page 9

By incorporating a nickel-chelating lipid (10% 1,2-di-(9Z-octadecenoyl)-sn-glycero-3-[(N-(5-amino-1-carboxypentyl)iminodiacetic acid)succinyl]) into our vesicles, our model recombinant platform construct bound tightly to the membrane surface irrespective of specific peptide-vesicle interactions.

Using this model ~~platform~~ construct, we dissected the contribution of the following physicochemical parameters: vesicle size, membrane charge, salt conditions and fluorophore identity (Figure 4, Figure S7).

Supporting this hypothesis, altering the pH of our sample solution led to a decrease in fluorescence anisotropy for the free state of ~~our recombinant platform~~ all recombinant constructs tested, potentially due to pH induced charge changes in the fluorescently-labeled C-terminal domain of our platform (Figure S9).

Page 10:

Analyzing this decay over time allows us to separate the depolarization due to the potentially faster rotational diffusion of the fluorophore from the slower diffusion of the platform construct or ~~platform construct~~/vesicle complex, providing mechanistic insights into the observed changes in steady-state anisotropy upon binding.

We again used the hexahistidine model ~~platform~~ construct (His₆-SUMO-fluorophore) to examine the bound-state behavior. We first attempted to fit the decay of the three states with a model corresponding to a single rotational correlation lifetime, consistent with a fluorophore rigidly attached to the platform construct (in a free state) or the ~~platform~~ construct/vesicle complex (in the bound states).

Page 11:

As a result, we explored using a model in which there is an additional hindered degree of freedom, representing a fluorophore that is non-rigidly attached to the platform construct.

As a result, early time anisotropy decay is governed primarily by the local motion of the fluorophore, likely arising from its attachment to the disordered C-terminal domain of our platform His₆-SUMO-fluorophore construct.

Page 12:

In the free state, there are significant contributions from both global and local motion of the platform construct and fluorophore, respectively.

4) While the authors claim that peptides where phenylalanine has been replaced with a tryptophan show the same membrane binding behavior as detected by their anisotropy-based method, it is to be expected that a tryptophan containing peptide binds more strongly to the bilayer than a phenylalanine containing peptide (see e.g. Whimley White hydrophobicity scale). This should be discussed.

This is an excellent point that we have unfortunately overlooked to address in the manuscript. The reviewer correctly points out that the hydrophilicities of the two tryptophan and phenylalanine are different and may lead to altered partitioning into the membrane bilayer. One of the most widely used hydrophobicity scales is the one introduced by Whimley and White, which lists tryptophan as slightly more hydrophobic than phenylalanine. There are however multiple other scales which rank amino acid hydrophobicity differently². Furthermore, when analyzing the peptide sequences used in the paper with and without the F->W mutation using Heliquest³, the overall hydrophobicity of the induced amphipathic helices do not change that much. Specifically, the WT hydrophobicity of Amphiphysin-N25 alpha helix is calculated to be 0.185, while the W containing mutant's is 0.203.

We included the following text on page 6 of the manuscript.

Conveniently, all peptides tested contain at least one phenylalanine, which can be mutated into a physiochemically similar tryptophan, enabling parallel measurements using fluorescence anisotropy (Figure S2) and tryptophan fluorescence (Figure S3). Although tryptophan has a higher hydrophobicity than phenylalanine, which is expected to somewhat alter the binding affinity of the inducible amphipathic helix, the overall effect should not yield a significantly different binding profile than the wildtype helix sequence^{13,34,35}.

Comments 5-8 are grouped together

5) The sentence “Taken together, the fluorescence decay data indicate the decrease in steady-state fluorescence anisotropy occurs because there is a transition” sounds a bit circular – perhaps it would help if the authors explained at this point what they mean by ‘transition’.

6) Next, I’m not sure what is meant by ‘meaningful contribution’. Would it be more accurate to replace ‘meaningful’ by ‘measurable’, ‘detectable’, or ‘significant’?

7) The fragment: “the overall decay is dominated by the local motion of the fluorophore

and there is a concomitant increase in local mobility” sounds like repeating the same thing twice (local motion and local mobility increase).

8) In the following sentence the authors then refer to “oscillatory motion” of the fluorophore. This does not seem to be physically correct as oscillatory motion is deterministic, whereas the authors are observing Brownian motion – a stochastic process.

The reviewer correctly points out an inconsistency in our writing that leads to confusion. To address these comments we have reconstructed that paragraph on page 12 as follows:

~~Taken together, the fluorescence decay data indicate the decrease in steady-state fluorescence anisotropy occurs because there is a transition. In the free state, there are meaningful contributions from both global and local motion of the platform and fluorophore, respectively. Upon binding, the overall decay is dominated by the local motion of the fluorophore and there is a concomitant increase in local mobility. We geometrically interpret this to represent a release of steric hindrance of the fluorophore. In the presence of a negatively charged membrane, the oscillatory local motion of the fluorophore increases due to the surface potential (an effect that does not occur with neutral membranes). The system's sensitivity to variations in membrane charge and ionic strength further supports this mechanistic interpretation.~~

Taken together, the fluorescence decay data indicate the decrease in the steady-state fluorescence anisotropy occurs because of shift in the contribution weights from the individual components of the deconvolved motion. In the free state, there are significant contributions from both global and local motion of the platform construct and fluorophore, respectively. Upon binding, the overall decay is overwhelmed by the local motion of the fluorophore relative to the slow motion of the bound recombinant platform. We interpret this as a release of steric hindrance of the fluorophore. In the presence of a negatively charged membrane, the stochastic motion local motion velocity of the fluorophore increases due to the surface potential (an effect that does not occur with neutral membranes, Figure 7c).

9) The value of the NMR experiment is questionable as membrane binding could not be detected with this approach. Perhaps it would help to motivate this experiment more clearly.

We agree that the current wording and explanation of the NMR data is lacking regarding its motivation. Overall, it is unfortunate that the membrane bound state could not be directly observed by the NMR data. However, we believe that there is value in presenting these results as they demonstrate a few key points that support our overall mechanistic interpretation. To accurately express the conclusions that can be extracted from our NMR experiments we have proceeded to rework the relevant paragraph on page 12 of the revised manuscript. Furthermore, we have separated the NMR results in a new figure (Fig 8) with added an AlphaFold3 model that can help visualize some conclusions.

To corroborate our mechanistic interpretation that fluorophore mobility increases in the bound state of the platform, we investigated the structural and dynamic properties of our

platform by nuclear magnetic resonance (NMR) spectroscopy. First, to verify that fluorophore interactions with the rest of the platform remained non-specific, we compared the NMR spectra of $^{13}\text{C}/^{15}\text{N}$ -labeled versions of our hexahistidine model platform with and without a fluorophore. Spectral differences predominantly localized to the C-terminal region where the fluorophore is conjugated (Figure S13), although some resonance signals within the SUMO domain were doubled indicating a non-negligible fluorophore proximity effect. Nevertheless, the small chemical shift differences between split resonances show that the structure of the folded SUMO domain remained virtually unchanged. Moreover, since specific SUMO-ligand interactions usually produce much larger shift changes³⁸ and resonance doubling was also observed near the C-terminal conjugation site, we deem any SUMO-fluorophore interaction to be unspecific. Next, we investigated the NMR response of the platform in the presence of vesicles aiming to gain insight into the dynamic behavior of our platform. While the large particle size of the vesicle-bound platform did not allow the direct observation of the bound state by NMR, the protein fraction remaining free in solution experienced a relaxation contribution from its exchange with the vesicle-bound state. This contribution diminished the signal intensities of the disordered C-terminal region and the rest of the platform domains to different degrees (Figure 5e) showing their dynamic, loose coupling. In the vesicle-bound state a loss of non-specific SUMO domain-fluorophore interactions and/or an altered proximity behavior may loosen this coupling even further. Overall, a lower steady state fluorescence anisotropy signal in the vesicle-bound relative to the free state is therefore compatible with an increase in the conformational freedom of the fluorescently-tagged C-terminal region of our platform upon vesicle binding.

To further analyze fluorophore mobility, we investigated the structural and dynamic properties of our platform by nuclear magnetic resonance (NMR) spectroscopy. First, we compared the NMR spectra of $^{13}\text{C}/^{15}\text{N}$ -labeled versions of our His₆-SUMO-fluorophore construct with and without fluorophore to assess its impact on the SUMO domain. For most residues, NMR resonances remained virtually unchanged in the presence of fluorophore (Figure 8a-b and Figure S10). However, for a few residues, resonances shifted while often maintaining a minor resonance at the original peak position (Figure 8a). Mass spectrometry showed a fluorophore labeling of ~78.6% (Figure S12), suggesting that the minor peaks represent unlabeled protein. In AlphaFold3 models of our construct, the SUMO secondary structure ends at R104 (Figure 1c) and the largest spectral differences localized to E105-C110 in accordance with the C-terminal linkage of the fluorophore⁴⁰. Within the SUMO domain, some chemical shift differences were detected (Figure 8b) but they are small relative to well-defined SUMO-ligand interactions⁴¹. Mapping these chemical shift changes on the SUMO structure (Figure 8c) produced no clear insight into specific contacts of the fluorophore with SUMO. With a net negative charge and conjugated ring system, the fluorophore likely makes differential contacts with the protein surface, but a clear preference was not discernible.

Next, we investigated NMR spectral changes of the His₆-SUMO-fluorophore construct in the presence of vesicles aiming to gain insight into the dynamic behavior of our platform. While the large particle size of the vesicle-bound platform construct did not allow the direct observation of the bound state by NMR, the protein fraction remaining free in solution

experienced a relaxation contribution from its exchange with the vesicle-bound state. This contribution diminished the signal intensities of our construct to different degrees (Figure 8d and Figure S11), documenting a loose coupling between the N-terminal region (M1-E32), the SUMO domain, and the C-terminal region (Q106-C110). These transitions correlate with low confidence scores of the N- and C-terminal regions in AlphaFold3 models (Figure 8c) and illustrate their dynamically unfolded nature. In the vesicle-bound state, this loose coupling is expected to be preserved but an altered chemical and dynamic environment may change any fluorophore-SUMO contacts and dynamic fluorophore-SUMO coupling. Accordingly, an increase in the conformational freedom of the fluorescently-tagged C-terminal region upon vesicle binding could explain the lower steady state fluorescence anisotropy signal in the vesicle-bound relative to the free state.

10) The authors write that the technique is not limited by an upper anisotropy. This is a bit misleading in two ways. First, any fluorescence anisotropy measurement is bound by a fundamental upper limit. What the authors mean to express is that their anisotropy decreases upon binding – but there is of course a lower limit to fluorescence anisotropies as well. Perhaps the authors could reword this section to explain more clearly what they have in mind.

This is an insightful point that we have overlooked in the original manuscript. To address this, we have removed that section from page 14.

By demonstrating that our recombinant platform can serve as a precise tool for measuring association with negatively charged membranes, this work greatly expands the scope of fluorescence anisotropy-based sensing, showcasing that a decrease in anisotropy can be a highly efficient biomolecular fluorescence probe. ~~In contrast with a traditional fluorescence anisotropy approach, this technique is not constrained by an upper anisotropy limit (easily reached by large proteins) rendering the free and bound states indistinguishable from one another.~~ Furthermore, our findings show that there is no inherent need for lengthy structural investigations for an “ideal” rigid fluorescent labeling site when designing anisotropy probes.

11) I’m not sure it is accurate to say that the high throughput nature of the technique ‘outweighs’ the technical limitation that high membrane charge and low ionic strength is required. Instead these are simply advantages and disadvantages of the technique that future users need to be aware of.

The reviewer is correct to point out that the advantages of the technique do not “outweigh” its limitations. We took steps to convey this sentiment in page 14 of the revised manuscript.

While the low salt conditions and high membrane charge concentrations used in this study do not fully replicate the biological milieu, we believe ~~the advantages of that~~ a controlled, high-throughput system ~~strongly outweigh these limitations.~~ can serve as an invaluable initial screening tool and streamline key parameters to be prioritized for rigorous testing in a more physiologically relevant setting. While the requirement for charged lipid membranes somewhat limits our system, the intrinsically high negative surface charge of key organelles

such as endosomes and mitochondria ensures that our conditions remain well-suited for capturing a broad range of biologically meaningful interactions. Most importantly, the ability of our platform to detect AH sensitivity to certain lipid species in a high throughput manner offers an invaluable new approach to uncovering the peptide code governing AH membrane binding. Discovering inducible amphipathic helix function and the parameters affecting its membrane association can allow us to distinguish between the protein domains responsible for binding from those involved in catalytic function, elucidating fundamental principles that govern membrane-associated biochemical pathways.

References

1. Robustelli, J. & Baumgart, T. Membrane partitioning and lipid selectivity of the N-terminal amphipathic H0 helices of endophilin isoforms. *Biochim. Biophys. Acta BBA - Biomembr.* **1863**, 183660 (2021).
2. Simm, S., Einloft, J., Mirus, O. & Schleiff, E. 50 years of amino acid hydrophobicity scales: revisiting the capacity for peptide classification. *Biol. Res.* **49**, 31 (2016).
3. Gautier, R., Douguet, D., Antony, B. & Drin, G. HELIQUEST: a web server to screen sequences with specific alpha-helical properties. *Bioinforma. Oxf. Engl.* **24**, 2101–2102 (2008).
4. Chen, Z., Zhu, C., Kuo, C. J., Robustelli, J. & Baumgart, T. The N-Terminal Amphipathic Helix of Endophilin Does Not Contribute to Its Molecular Curvature Generation Capacity. *J. Am. Chem. Soc.* **138**, 14616–14622 (2016).

Reviewer comments are in bold.

Author responses are in regular font.

Removed text from the original manuscript is in red.

All revisions in manuscript are in blue.

Reviewer #1

In the revised manuscript, the authors are more transparent in acknowledging the requirement for low salt concentration and negatively charged lipids in their experimental platform. However, this phenomenon is not mentioned in the abstract, and the authors still try to minimize its significance, postulating in their rebuttal to me: "there is likely a secondary mechanism by which the decreases in anisotropy are observed upon binding that are not dependent on the charge of the vesicle surface. This observation was key in proposing a potential steric hindrance between the N-terminus of the protein and the fluorophore."

To address this concern, we have added the following in the abstract:

By using fluorescence anisotropy decay measurements, solution NMR experiments, and solution FRET, we deduce that this phenomenon likely occurs due to the local increase in fluorophore motion upon binding to the membrane enabled by vesicle membrane charge under low-salt conditions.

The above statement by the authors is faulty because all of the decreases in anisotropy that are observed upon vesicle binding do in fact depend on the presence of negative charges on the vesicle surface. The authors cite Figures 5b, 7b and S6a as their counter evidence. (Actually, Figure S6a is not evidence of this phenomenon because binding in this figure occurs via His-tag-nickel ion interaction. I think the authors mean Figure S6b.) In all of these figures, they wrongly assert that the lipid mixture of DOPC/DGS-NTA(Ni) is neutral. However, this is incorrect. While NTA has a net charge of -2 from an amine group and three carboxylates that appear to cancel out the positive charge from Ni²⁺, binding of Ni²⁺ actually displaces a proton from the amine group, so DGS-NTA(Ni) has in fact a -1 charge. Thus, the authors mistakenly believe that these observations challenge the need for an electrostatic interaction to observe fluorescence anisotropy changes; however, this is not the case.

We thank the reviewer for pointing out this oversight in our manuscript. We have revised the manuscript and changed all mentions of "uncharged" or "neutral" vesicles referring to DOPC/DGS-NTA vesicles to "low membrane charge". The changes are as follows:

Page 10:

To ensure binding occurs ~~under electrostatically neutral~~ low membrane charge conditions, microscopy measurements confirmed colocalization occurs with ~~electrostatically neutral~~ DOPC/DGS-NTA(Ni) vesicles (Figure S6).

Page 11:

These experiments required decay measurements in three states: free in solution, bound to highly negatively charged vesicles (to recapitulate the observed steady-state effect), and bound to vesicles with low charge (which exhibits no change in anisotropy even though the protein is membrane bound) ~~which serves as a control, as it does not exhibit a decrease in anisotropy and thus reveals baseline behavior~~.

Also found in labels within Figure 6 (Previously Figure 7).

The "steric hindrance" mechanism between the N-terminal tail of the protein and the fluorophore arises from a misinterpretation of the NMR data presented in Figure 8, a point I made in my initial review. A cursory glance at the structures shown in Figures 7 and 8 show that this is impossible, with the N-terminal tail and the fluorophore-labeled C-terminal tail pointing away from opposite ends of the SUMO domain. The model structure presented in Figure 7d is thus impossible.

We have removed Figure 7d from the manuscript completely to avoid confusion.

The NMR data in Figure 8 is misinterpreted by supposing that because the N- and C-terminal tails show similar reduction in intensity induced by vesicle binding, they must be somehow close to each other or interacting. However, this is entirely not the case, all you can conclude is that the N-terminal tail is flexible and so is the C-terminal tail. In fact, this phenomenon is better explained by the SUMO domain itself forming transient electrostatic interactions with the negatively charged vesicles, with increasing mobility observed in the tails as one moves further away from the SUMO domain. In contrast, if only the N-terminal tail were tethered to the vesicles as suggested by the authors in Figure 7, one would not expect to see such restriction in the proximal C-terminal tail.

We have also removed parts of the introduction where we commented on the C-terminal tail. Furthermore, we have added several sentences to both introduce new experiments we performed for clarifying the role of electrostatic interactions on the underlying phenomenon, as well as to present a clear picture on how these tie together with previous experiments. The changes are as follows:

Pages 4 and 5:

To illuminate the mechanism that leads to this phenomenon, we investigated the dynamic structure of our recombinant platform using a combination of fluorescence anisotropy decay, nuclear magnetic resonance measurements and solution fluorescence resonance energy transfer (FRET) measurements. These techniques reveal the presence of a secondary motion (stemming from the disordered nature of the fluorescently-tagged C-terminal domain) that conformationally relaxes and allows for an increase in the local mobility of the fluorophore upon membrane binding. This increase in local mobility dominates the overall rotational motion of the bound construct/vesicle system, leading to an observed reduction in the steady state fluorescence anisotropy. of secondary, charge induced interactions between part of the folded domain of SUMO and the negatively charged fluorophore that affect the conformation of the system both in the free and vesicle bound states. Upon vesicle binding, the fluorescent label experiences varying degrees of secondary conformational freedom dependent on the amount of “visible” membrane charge, yielding anisotropies lower than those of the free state. When membrane charge decreases, or the environmental salt concentration increases, the dynamic range in fluorescence anisotropy (i.e. the difference between the free and fully bound states) is reduced, limiting the accuracy at which binding efficiencies can be determined. To assess whether there are electrostatically-mediated transient interactions between the folded SUMO domain, vesicles, and fluorophore, we introduced a series of charge-altering mutations to the construct, minimizing electrostatic interactions and thereby increasing the dynamic range in elevated salt conditions, expanding the platform’s potential application to the near physiological salt regime for negatively charged membranes or to near uncharged membranes in the low salt regime.

On page 13 we have reframed parts of the interpretation of the anisotropy decay data:

Upon binding, the overall decay is overwhelmed by the local motion of the fluorophore relative to the slow motion of the bound recombinant platform. ~~We interpret this as a release of steric~~

~~hindrance of the fluorophore~~. This effect is present in the bound state of vesicles with both high and low negative membrane charge (Figure 6a-c). A detectable decrease in steady-state anisotropy relative to the free state is only observed when the amount of negative membrane charge “visible” by the fluorophore exceeds a threshold, rather than resulting in a simple “cancellation” of the anisotropy increase induced by vesicle binding. This threshold is substantially reduced upon introduction of the arginine mutations (Figure S15), showcasing that neutralization of positive charges from proximal arginine residues greatly enhances the local flexibility of the fluorophore, promoting motion that is increasingly decoupled from the protein construct-vesicle system. While it is unclear if this phenomenon is directly responsible for the decrease in anisotropy of the original constructs, it suggests an overall molecular mechanism that locally restrains the fluorophore in the free state but is released in the presence of a negatively charged membrane. As a result, the steady-state anisotropy becomes lower than that of the unbound state; while coincidental in magnitude, this effect is highly robust and reproducible, enabling its use as a molecular reporter of binding. ~~In the presence of a highly negatively charged membrane, the stochastic motion local motion velocity of the fluorophore increases due to the surface potential (an effect that does not occur with neutral membranes, Figure 7c).~~

Finally, to address reviewer concerns about interpretation of the NMR data, we have removed the following text on page 15:

These transitions correlate with low confidence scores of the N- and C-terminal regions in AlphaFold3 models (Figure 7c) and illustrate their dynamically unfolded nature. In the vesicle-bound state, this loose coupling is expected to be preserved but an altered chemical and dynamic environment may change any fluorophore-SUMO contacts and dynamic fluorophore-SUMO coupling. ~~Accordingly, an increase in the conformational freedom of the fluorescently-tagged C-terminal region upon vesicle binding could explain the lower steady state fluorescence anisotropy signal in the vesicle bound relative to the free state.~~

In conclusion, the observed fluorescence anisotropy change is very dependent on electrostatic forces that initially restrict the motion of the fluorophore via interaction of the negatively charged fluorophore-C-terminal tail with a limited positively charged patch on the SUMO domain (one can see it is limited from the authors' new AlphaFold3 model). The fluorophore is released by electrostatic interaction between the SUMO domain and negatively charged vesicles. However, this electrostatic interaction is not strong enough on its own and requires additional binding of N-terminal vesicle-binding amphipathic helices that the authors are proposing to study.

To directly address the reviewer's concerns, we designed a series of experiments that uncovered important new insights into our phenomena. To assess the effect of the positively charged arginine patch on the fluorophore, we introduced arginine-to-alanine mutations into our His₆-SUMO construct. We expressed and purified two variants: a double mutant (R75A/R82A, denoted 2RXA) and a quadruple mutant (R57A/R75A/R82A/R104A, denoted 4RXA). Circular dichroism measurements confirmed that the structural integrity of the folded SUMO domain was preserved in the mutants compared to the original (Figure S14).

We then repeated the experiments from Figure 5 using both the new mutants and the original construct (Figure S15). Strikingly, we observed a significant decrease in anisotropy upon binding to low-charge membrane vesicles (Figure S15a-c) and under 50 mM and 100 mM NaCl conditions (Figure S15d-f) for the mutants relative to the original construct. This result indicates that the fluorophore in the mutant constructs exhibits greater conformational flexibility in the bound state over the free state, than the original construct. We conclude that the arginine patch restricts the fluorophore's motion through transient electrostatic interactions. However, these interactions are not the primary driver of the anisotropy decrease, since removing that arginine patch does not eliminate the decrease in anisotropy, rather it greatly enhances it.

To further investigate how this transient electrostatic interaction is affected by the close proximity of the negatively charged membrane, we performed FRET measurements using vesicles doped with a fluorescent acceptor (headgroup conjugated Rhodamine-B-DOPE) compatible with the OG488 protein conjugated donor. Fluorescence lifetime decays were collected under varying membrane charge (Figure S16) and NaCl concentration (Figure S17), and mean lifetimes were used to estimate FRET efficiencies. This histogram-based lifetime analysis was chosen because vesicle binding produces highly heterogeneous, multi-exponential decays that cannot be reliably fit with simple models. Control constructs lacking membrane binding exhibited single-exponential decays (~4 ns, consistent literature for OG488), whereas membrane-bound constructs showed complex decays consistent with multiple conformational states.

We observed a robust increase in FRET efficiency with increasing DOPS content for all His₆-SUMO constructs, with the effect saturating at ~30–45% DOPS (Figure S16m-l). This trend indicates that higher membrane charge stabilizes conformations in which the donor fluorophore is positioned closer to the membrane surface. However, the moderate saturation values (FRET efficiency ~0.3) suggest that the fluorophore does not reside directly at the membrane interface. Arginine-to-alanine mutations preserved the overall dependence on membrane charge but

progressively reduced the magnitude of the FRET increase, pointing to a key role for electrostatic interactions in governing the fluorophore's conformations.

To directly test the electrostatic origin of these effects, we performed complementary FRET experiments at fixed membrane composition while varying NaCl concentration (Figure S17). Increasing NaCl presence led to a systematic reduction in FRET efficiency, with no detectable FRET at 300 mM NaCl, a result likely happening due to a more extended protein conformation rather than loss of membrane binding. Progressive arginine-to-alanine mutations within the SUMO arginine patch further diminished FRET across the same salt range, supporting a model in which these residues contribute into restricting the fluorophore in the membrane bound states. Independent steady-state fluorescence anisotropy measurements of the free protein state corroborated this interpretation (Figure S18), revealing lower anisotropy for mutant constructs even in the free state regardless of NaCl concentration, indicative of increased local fluorophore mobility and weaker coupling to the overall protein conformation.

Overall, these data highlight the importance of the charged arginine patch in the fluorophore's conformational freedom. While the presence of the patch is not a direct contributor to the emergence of the anisotropy decrease effect (since the phenomenon gets stronger by removing it), it was a significant limiting factor hindering our system's response in the near-physiological regime. Although we cannot precisely elucidate the mechanism by which anisotropy decrease upon binding, our new data suggest that electrostatics does indeed play an important role. We have outlined these new experiments and insights in the following parts of the original manuscript:

On pages 15-18:

Electrostatic interactions affect fluorophore mobility

The presence of a small positively charged patch on the surface of the SUMO folded domain (Fig 7c) raised the question whether transient electrostatic interactions between this patch with the negatively charged fluorophore and/or membrane surface could contribute to the anisotropy decrease phenomenon. To address this, we expressed and purified two mutant His₆-SUMO-fluorophore constructs: a double mutant (R75A/R82A, dubbed 2RXA) and a quadruple mutant (R57A/R75A/R82A/R104A, dubbed 4RXA). The CD spectra of the mutated constructs showed no apparent changes in protein structure (Figure S14). We measured the steady-state fluorescence anisotropy binding curves against increasing membrane DOPS content and environmental NaCl concentration, mirroring measurements from Figure 5 (Figure S15). Remarkably, both mutants continued to exhibit a decrease in anisotropy upon binding,

while substantially expanding the sensitivity of the platform. Furthermore, the 4RXA variant exhibiting pronounced anisotropy decreases even with minimally charged membranes and under near-physiological salt conditions. These results indicate that the charged arginine patch may electrostatically restrict fluorophore mobility in the membrane-bound state, thereby limiting the extent of anisotropy reduction achievable under low-charge or high-salt conditions. However, the underlying mechanism remains uncertain, as the data do not distinguish whether this effect arises solely from fluorophore–protein electrostatics or from transient SUMO–membrane interactions that affect the overall binding conformation.

To further probe the steric conformation of membrane-bound His₆-SUMO model constructs, we performed solution FRET measurements using vesicles doped with fluorescently labeled DOPE (headgroup-conjugated Rhodamine-B) as FRET acceptors for the OG488 protein label. Donor fluorescent decays were acquired as described in Methods under varying membrane DOPS content and NaCl concentration (Figure S16). Mean fluorescence lifetimes were extracted from these decays and used to estimate FRET efficiencies. We adopted this decay histogram-based analysis rather than direct fitting of the decay profiles because the vesicle-bound protein system produces highly complex FRET-induced multi-exponential decays, making it difficult to reliably resolve the underlying distribution of contributing decay components. The estimated FRET efficiency was therefore used as an inverse proxy for the mean donor–acceptor distance, and by extension, the average separation between the N-terminally conjugated donor fluorophore and the membrane surface. For all vesicle-containing samples, lipid-to-protein molar ratios were kept sufficiently high to ensure that the majority of protein was membrane-bound. In the absence of any vesicle interactions (such as for the hexahistidine lacking “CONT” constructs), donor decays were well described by a single exponential with a mean lifetime of ~4ns (consistent with literature for OG488⁴²), whereas

membrane binding resulted in highly heterogeneous, multi-exponential decays, reflecting the diverse conformational states adopted by the protein constructs in the membrane-bound state. We observe a consistent increase in FRET efficiency for His₆-SUMO constructs bound to vesicles with increasing DOPS membrane content, a trend that is conserved across all mutant constructs tested and that tends to plateau at approximately 30–45% DOPS (Figure S16k-m). This behavior indicates that increasing membrane charge promotes vesicle-bound conformational states in which the donor fluorophore resides closer to the membrane surface. However, because the FRET efficiencies saturate at values around ~0.3 (Figure S16k-m), these conformations are unlikely to position the fluorophore directly at the membrane interface, which would be expected to yield substantially higher FRET efficiencies. Arginine-to-alanine substitutions within the SUMO domain preserve the overall dependence of FRET efficiency on membrane charge but progressively attenuate the magnitude of the increase, suggesting the conformational behavior of the fluorophore on the membrane surface is heavily dependent on electrostatics.

To further assess whether electrostatic interactions underlie these conformational changes, we performed analogous FRET experiments in which the membrane composition was held constant while the NaCl concentration was systematically varied (Figure S17). Increasing the NaCl concentration from 0 to 100 mM resulted in a consistent decrease in FRET efficiency. At 300 mM NaCl, no detectable FRET signal was observed. The lack of FRET in this case likely results from a relaxed protein state, positioning the C-terminal fluorophore beyond the FRET radius, rather than a lack of binding (hexahistidine-tagged proteins are often captured on nickel columns at 500 mM NaCl). Consistent with this interpretation, progressive mutation of arginine residues (R57/R75 and R57/R75/R82/R104) increasingly diminished FRET efficiencies across the same NaCl concentration range, indicating a more relaxed fluorophore state residing further from the membrane surface. Steady state fluorescence anisotropy measurements

corroborate this phenomenon, showcasing a distinct lower fluorescence anisotropy for mutant constructs regardless of NaCl concentration (Figure S18a-b) in the free state. This lower fluorescence anisotropy suggests local enhanced mobility of the fluorophore that is less coupled to the mobility of the construct itself.

Taken together, these measurements suggest a composite mechanism where the anisotropy is modulated by both the global positioning of the protein relative to the membrane and the local steric freedom of the fluorophore. While salt concentration affects the average separation distance between the SUMO domain and the lipid bilayer by screening electrostatic interactions, the proximal arginine cluster also acts as an electrostatic tether that restricts the C-terminal conjugated fluorophore. Removal of this positive patch decouples the fluorophore from the protein core, resulting in enhanced local mobility, thereby potentially extending the utility of this platform to the near-physiological regime.

Finally, we have changed the following on page 17:

While the low salt conditions and high membrane charge ~~contents~~ ~~concentrations~~ used in this study do not fully replicate the biological milieu, we believe that a controlled, high-throughput system can serve as an invaluable initial screening tool and streamline key parameters to be prioritized for rigorous testing in a more physiologically relevant setting. Although ~~While~~ the requirement for charged lipid membranes somewhat limits our system, the intrinsically high negative surface charge of key organelles such as endosomes and mitochondria ensures that our conditions remain well-suited for capturing a broad range of biologically meaningful interactions. ~~Furthermore, the enhanced anisotropy-decrease responses exhibited by the mutated versions of our model construct highlight opportunities to refine the platform and improve its relevance under near-physiological regime.~~ Most importantly, the ability of our platform to detect AH sensitivity to certain lipid species in a high throughput manner offers an invaluable new approach to uncovering the peptide code governing AH membrane binding.

We have further updated the data in figure 4 for consistency with our new measurements, without having any difference in overall results or interpretation. We moved Figure 6 into the supplement (Now Figure S8) to better align with the figure separation introduced by the new supplemental data. Finally, a few citations have been updated as well as the Methods.

Reviewer #2

The authors have thoughtfully and successfully addressed my comments and concerns and have significantly revised their manuscript text.

I just have one more comment: it seems that the other referee has suggested an intriguing, specific, alternative mechanism. It may be worthwhile for the authors to (re-consider) the possibility of being more specific, when they suggest - in the revised manuscript text - that alternative mechanisms are possible.

As mentioned above, we have proceeded to alter several parts of the text to address the comments by reviewer 1. We have also performed extensive experiments trying to directly elucidate the mechanism noted by the reviewer and try to explain its effects on our system.